# Reference compounds for characterizing cellular injury in high-content cellular morphology assays

Jayme L. Dahlin [1,2,13] ✉, Bruce K. Hua [2,13], Beth E. Zucconi[3],
Shawn D. Nelson Jr[4], Shantanu Singh [5], Anne E. Carpenter [5],
Jonathan H. Shrimp [6], Evelyne Lima-Fernandes[7], Mathias J. Wawer [2],
Lawrence P. W. Chung[2], Ayushi Agrawal [2], Mary O'Reilly[8],
Dalia Barsyte-Lovejoy [7], Magdalena Szewczyk[7], Fengling Li[7], Parnian Lak[9],
Matthew Cuellar [10], Philip A. Cole [3], Jordan L. Meier [6], Tim Thomas [11],
Jonathan B. Baell [12], Peter J. Brown [7], Michael A. Walters [10],
Paul A. Clemons [2], Stuart L. Schreiber[2] & Bridget K. Wagner [2] ✉

Robust, generalizable approaches to identify compounds efficiently with undesirable mechanisms of action in complex cellular assays remain elusive. Such a process would be useful for hit triage during high-throughput screening and, ultimately, predictive toxicology during drug development. Here we generate cell painting and cellular health profiles for 218 prototypical cytotoxic and nuisance compounds in U-2 OS cells in a concentration-response format. A diversity of compounds that cause cellular damage produces bioactive cell painting morphologies, including cytoskeletal poisons, genotoxins, non-specific electrophiles, and redox-active compounds. Further, we show that lower quality lysine acetyltransferase inhibitors and nonspecific electrophiles can be distinguished from more selective counterparts. We propose that the purposeful inclusion of cytotoxic and nuisance reference compounds such as those profiled in this resource will help with assay optimization and compound prioritization in complex cellular assays like cell painting.

Cellular nuisance compounds are a significant burden in high-throughput screening (HTS), high-content screening (HCS), and chemical biology. These compounds can appear to be bioactive yet act through nonspecific and poorly optimizable mechanisms of action (MoA) such as redox cycling, nonselective reactivity, and cytotoxicity[1,2]. Compounds causing cellular damage by more specific MoAs (e.g., tubulin and electron-transport chain poisons) can also be undesirable in certain contexts. Thus, cell-active compound prioritization can be difficult due to the uncertainty regarding the mechanism(s) producing phenotypic readouts[3]. Cell-free assays for

[1]National Center for Advancing Translational Sciences, National Institutes of Health, Rockville, MD, USA. [2]Chemical Biology and Therapeutics Science Program, Broad Institute, Cambridge, MA, USA. [3]Division of Genetics, Departments of Medicine and Biological Chemistry and Molecular Pharmacology, Harvard Medical School and Brigham and Women's Hospital, Boston, MA, USA. [4]Stanley Center, Broad Institute, Cambridge, MA, USA. [5]Imaging Platform, Broad Institute, Cambridge, MA, USA. [6]Chemical Biology Laboratory, Center for Cancer Research, National Cancer Institute, National Institutes of Health, Frederick, MD, USA. [7]Structural Genomics Consortium, University of Toronto, Toronto, ON, Canada. [8]Pattern, Broad Institute, Cambridge, MA, USA. [9]Department of Pharmaceutical Chemistry and Quantitative Biology Institute, University of California San Francisco, San Francisco, CA, USA. [10]Institute for Therapeutics Discovery and Development, University of Minnesota, Minneapolis, MN, USA. [11]Department of Medical Biology, University of Melbourne, Parkville, VIC, Australia. [12]Medicinal Chemistry Theme, Monash Institute of Pharmaceutical Sciences, Monash University, Parkville, VIC, Australia. [13]These authors contributed equally: Jayme L. Dahlin, Bruce K. Hua. ✉e-mail: jaymedahlin@gmail.com; bwagner@broadinstitute.org

compound-dependent interferences are helpful, but they may not model the ideal conditions for compound-mediated interference, such as xenobiotic metabolism or specific cellular microenvironments[4,5]. Instead, investigating cell-active compound MoA usually requires resource-intensive cellular assays that may also need extensive customization. Therefore, simple tools and resources, applicable to many areas of biology, which would help prioritize chemicals in cell-based assays without requiring extensive counter-screening are needed.

We previously participated in the development of an unbiased, multiplexed high-content cellular morphology assay ("Cell Painting", CP), which labels DNA, ER, nucleoli, cytoplasmic RNA, F-actin, Golgi apparatus, plasma membrane, and mitochondria[6–8]. The CP assay has been used for its strong information content while being higher in throughput and lower in cost relative to other profiling techniques like transcriptomics[9]. Mechanistic hypotheses can be inferred from CP data when compounds share similar phenotypic profiles[10–13]. Many groups have used CP to biologically annotate novel synthetic and other chemical libraries[14–18].

An acknowledged and challenging aspect of HTS is that many cell-active compounds that emerge as hits result from undesirable MoAs[19]. Effective triage of such compounds typically involves an extensive post-screening cascade of secondary and tertiary assays. Despite widespread use of HCS, there are still many unanswered questions about the best practices and limitations of these assays, including CP, as it pertains to addressing undesirable as well as cytotoxic/cytostatic chemicals. The US Environmental Protection Agency used image-based profiles from the CP assay to characterize selected environmental chemicals' bioactivity and toxicity, compared to more expensive chemical safety assessments[18] and PROTACs have been tested for mitotoxicity using cell painting[20]. Groups have developed customized high-content assays to detect nephrotoxicity, pulmonotoxicity, antibiotic toxicity in mammalian cells, and other toxicities[21–23]. We previously developed two cell health assays using specific stains and antibodies and found, using CRISPR reagents targeting various cancer-related genes, that many of their outcomes can be predicted using the information in CP images[24]. None of these assays have been tested in the context of a broad set of cytotoxic and nuisance compounds.

We have therefore profiled a series of prototypical cytotoxic and nuisance compounds by the established CP assay to systematically characterize outcomes associated with compound-dependent cellular injury. These results demonstrate the utility of this approach in distinguishing low- from high-quality compounds, and provide a blueprint for routinely detecting nuisance compounds in triage activities during HTS.

## Results

### Characterization of cellular injury using cell painting

The relationship between cellular injury and cell-painting (CP) phenotypes was first examined by analyzing a published dataset of CP images. Retrospective analysis of public data from 30,616 compounds profiled by CP at 10 μM concentrations as part of the NIH Molecular Libraries Initiative (MLI) showed that wells with low cell counts tend to have strong phenotypes in the CP assay as measured by Mahalanobis distances (Fig. 1a)[25]. Similar trends have also been described previously[17].

Given this relationship between cellular health and bioactive morphology, we then independently performed CP on 218 cytotoxins and prototypical nuisance compounds in quantitative HTS (qHTS, or concentration-response) format with a typical concentration range of 0.6 to 20 μM[26]. The extracted morphological features were subjected to feature reduction, unsupervised hierarchical clustering, and principal component analysis (PCA). Notably, cell features directly and indirectly based on cell numbers were excluded from these analyses. Compounds associated with several cellular injury mechanisms produced distinct morphological clusters (e.g., tubulin poisons [cluster 8],

genotoxins [cluster 6]; Fig. 1b, c; Supplementary Fig. 1). Other classes were less active, possibly because nonspecific activity may occur only at concentrations higher than the 20 μM maximum concentration profiled here (i.e., tannins, saponins). The cluster with the most variance and occupying the largest area in the PCA plot was associated with a diversity of compounds causing gross cellular injury such as nonspecific electrophiles ("NSEs"), proteasome inhibitors, and miscellaneous cytotoxins (cluster 9, "gross injury"; Fig. 1b, c) that could not otherwise be assigned to a specific cytotoxic MoA. Similar to the MLI dataset, CP activity score and clustering were inversely correlated with cell number in this panel of cytotoxic compounds (Fig. 1d), even more so than in the diverse panels of small molecules previously studied.

Analyses were performed to estimate the generalizability and reproducibility of the cellular injury CP phenotypes. First, we analyzed the existing MLI dataset for correlation to clusters 1–9 from our independent profiling of cytotoxins and nuisance compounds, using the shared extracted features between the two datasets (Fig. 1e). We then prospectively re-tested 285 compounds from the MLI dataset with either high (MLI-HC) or no correlation (MLI-NC) to the gross injury phenotype (cluster 9; Supplementary Fig. 2)[25]. We found that, upon retesting, 98/119 (82%) and 21/166 (13%) of MLI-HC and MLI-NC compounds were called bioactive upon retesting based on Mahalanobis distances (Fig. 1e). Second, the performance of select cellular injury compounds and CP controls was assessed in independent experiments. The CP phenotypes were robust across independent experiments (mean correlation = 0.87 ± 0.06; Supplementary Fig. 3). These results indicate that CP phenotypes associated with cellular injury are reproducible and can be analyzed using historical datasets. Furthermore, the data suggest that characterizing the phenotypic signatures of nuisance and cellular injury compounds could be used to alert scientists about potential compound liabilities of HCS-bioactive compounds.

Closer inspection of individual compound profiles corroborated these general observations regarding CP activity and cell injury. For prototypical compounds, there was a general trend of increased CP activity scores and lower relative cell counts with increasing compound concentrations (Fig. 1f). The images themselves demonstrate the wide variety of morphologies exhibited by cellular injury compounds. For compounds such as staurosporine (7) and gliotoxin (117), the higher CP activity scores at higher compound concentrations coincided with changes in the assigned phenotypic cluster, with higher activities tending to be assigned to cluster 9. These data demonstrate that prototypical cytotoxic compounds produce significant and diverse CP-defined phenotypes.

### Electrophiles and cell-painting phenotypes

There is a growing interest in targeted electrophiles (TEs) in drug discovery and chemical biology[27]. A common concern of electrophiles is cellular toxicity. We therefore profiled a series of NSEs, inactive analogs (NSE-IAs), and 13 high-quality TEs targeting a variety of proteins (e.g., BTK, EGFR, FGFR, KRAS G12C)[27]. The goal of this profiling was to determine if there are differences in CP phenotypes between specific/optimized versus nonspecific electrophiles that could be distinguished with the aid of profiling cellular injury reference compounds.

Amongst this electrophile-focused subset, CP activity was also inversely correlated with cell number after compound treatment (Fig. 2a). Less/nonreactive analogs (NSE-IA) were generally inactive in CP and did not affect cell number. By contrast, NSEs and some TEs tended to decrease relative cell number and produce significant CP activity scores when tested at higher compound concentrations. PCA revealed that most of the NSEs and some TEs occupied the gross cell injury feature spaces, but only at compound concentrations ≥10 μM, whereas most NSE-IAs were inactive (Fig. 2a). Notably, some of the TE

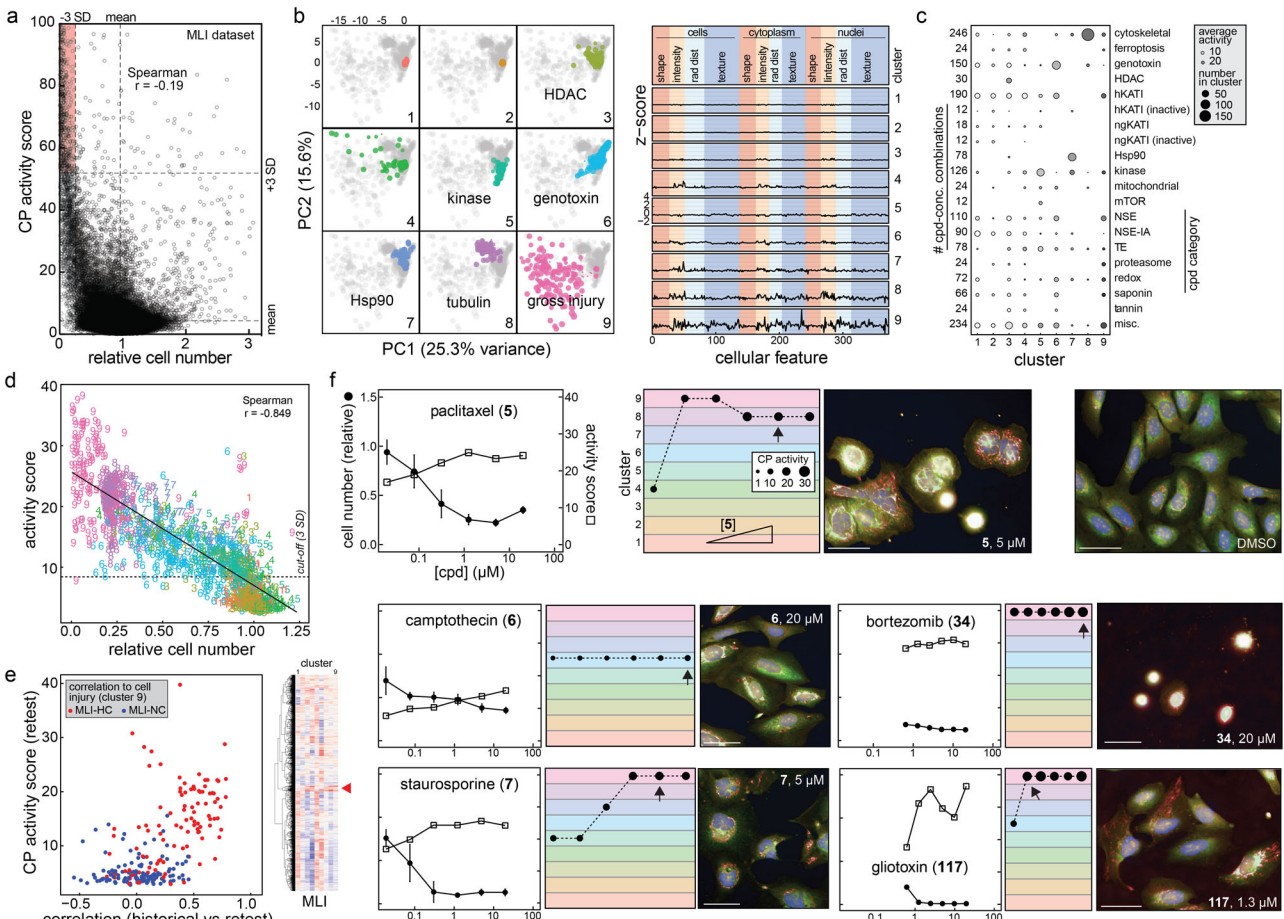

**Fig. 1 | Characterization of cell injury compounds using cell painting.** Compounds associated with cell injury were each profiled by cell painting (CP) after 24 h of compound exposure in U-2 OS cells. **a** Active CP compounds (by Mahalanobis distance) are enriched for decreased cell number in the Molecular Libraries Initiative (MLI) dataset[25]. Red highlight: compounds with the highest CP activity generally have the lowest relative cell numbers. **b** Cell injury compounds cause distinct CP phenotypes. Left: PCA plots showing unsupervised hierarchical clustering of CP phenotypes into nine clusters (some annotated when grossly associated with a compound category). Compounds causing gross cellular injury that could not be grouped into a more distinct MoA occupied with the largest area in the PCA plot. Right: reduced feature summaries for each cluster (all compound concentrations); rad dist, radial distribution. **c** "Dot plot" summary of cell injury compound categories by each CP cluster. The dot locations denote cluster identity, the dot sizes denote abundance within each compound subset, and the colors denote average activity score for all compounds in the subset. **d** The CP clusters of cell injury compounds correlate with cell number. Cluster 9, which is most

associated with gross cell injury, has the highest activity score and lowest relative cell number. The cut-off is 3 SD from the mean of DMSO-treated wells using the CP activity (Mahalanobis distances). **e** Compounds from the MLI dataset with high correlation (MLI-HC) to gross injury signature (cluster 9) are active upon re-testing (red), whereas compounds with no correlation to cluster 9 were not active upon re-testing (blue). Inset: heatmap and dendrogram shows pairwise correlation coefficients between each MLI CP compound profile and each of the nine clusters (red arrowhead, enrichment of cluster 9). **f** Select CP profiles of cellular injury compounds, demonstrating several different CP phenotypes of cell injury compounds. Rainbow plots denote assigned cluster at each compound concentration; arrow indicates compound concentration of representative image. For rainbow plots, note that phenotypic trajectories do not have to progress through each cluster before reaching the cluster 9 "gross injury" phenotype (dotted lines). Image scales: 50 μm. Data are mean ± SD of four intra-run technical replicates each performed on separate microplates. Source data are provided as a Source Data file.

targets are absent (KRAS G12C) or not highly expressed (BTK) in the profiled U-2 OS cells. Many NSEs (13/18, 72%), a subset of TEs (3/13, 23%), and only one NSE-IA (1/14, 7%) occupied the gross injury cluster 9 at 20 μM compound concentrations (Fig. 2a). Examination of the CP feature signatures showed gross similarities between NSEs and the cluster 9 (cell injury) signature, especially at higher compound concentrations (Fig. 2b).

Inspection of individual electrophile profiles demonstrated corroboration with these overall trends, where electrophile clustering tended to migrate towards the gross injury cluster 9 at higher compound concentrations (Fig. 2c). Most TEs did not produce the gross injury phenotype until concentrations in great excess of their expected EC$_{50}$ values. As with other cellular injury compounds, there was a general trend of increased CP activity scores and lower relative cell counts with increasing compound concentrations. These data

show that both NSEs and TEs can produce CP phenotypes associated with cellular injury, and that cellular injury phenotypes may help to identify highly reactive compounds, including TEs with off-target toxicities.

## Quality of KAT inhibitors are distinguishable by cell painting
In chemical biology, it is important to distinguish low- and high-quality chemical probes as early as possible to avoid wasting scientific resources and making erroneous conclusions. The utility of certain compound classes, including lysine acetyltransferase (KAT) inhibitors, has recently been questioned[28–31]. KATs are crucial components of eukaryotic DNA repair and nucleosome assembly, and aberrancies in histone acetylation and KAT function have been implicated in human pathologies including many cancers[32,33]. Numerous small-molecule KAT inhibitors have been reported[34], but many of these "historical"

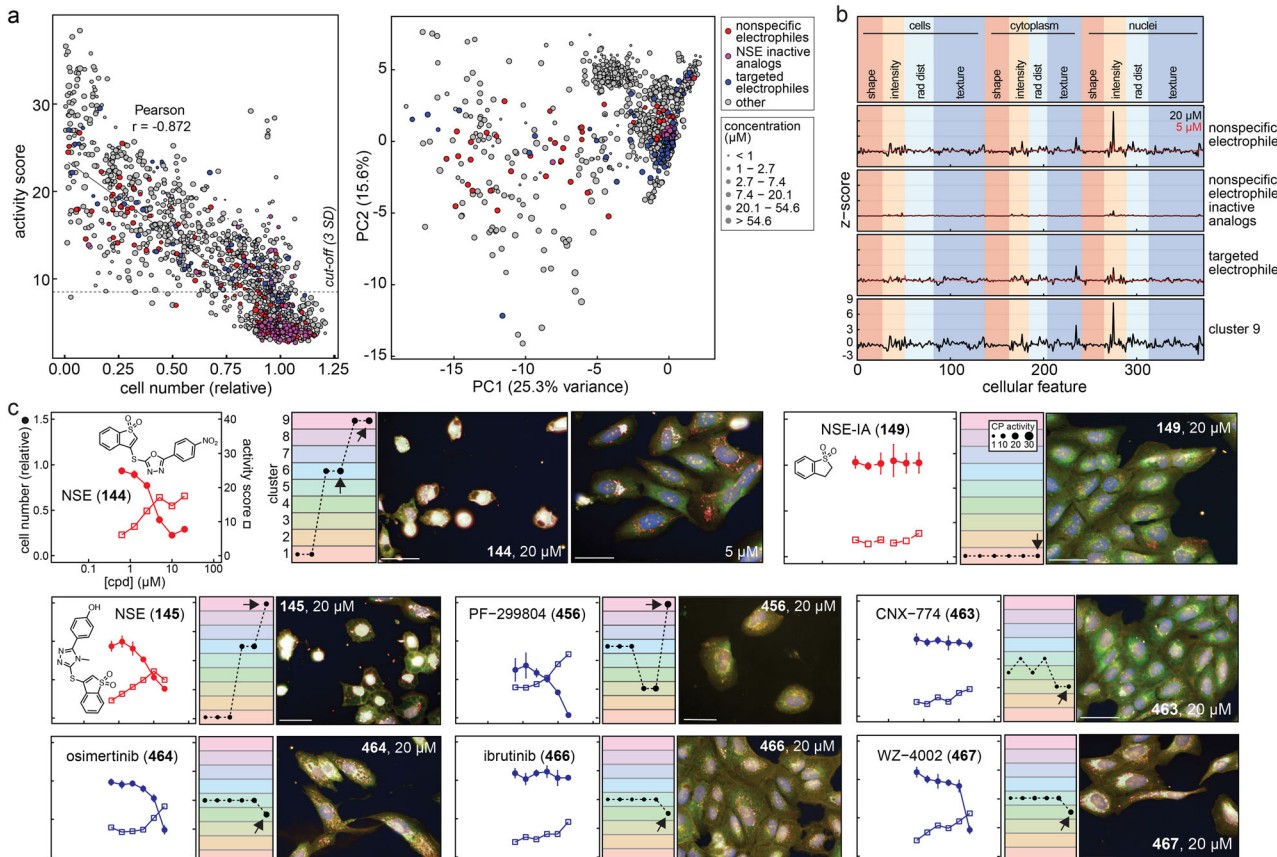

**Fig. 2 | Nonspecific electrophiles and select targeted electrophiles produce cellular injury phenotypes in cell painting.** A selection of electrophilic compounds were each profiled by cell painting (CP) after 24 h of compound exposure in U-2 OS cells. **a** Nonspecific electrophiles (NSEs) and some targeted electrophiles (TEs) perturb cell number, are scored as bioactive in CP, and occupy gross injury feature-spaces. The cut-off is 3 SD from the mean of DMSO-treated wells using the CP activity (Mahalanobis distances). **b** Reduced CP feature summaries for NSEs and TEs. NSE, but not inactive analogs, cause CP phenotypes similar to the gross cell injury phenotype, especially at higher compound concentrations. Targeted

electrophiles do not cause as pronounced CP phenotypes as NSEs. **c** Select CP profiles of NSEs and TEs. The NSE **144**, but not the inactive analog **149**, causes a cell injury CP phenotype at the higher concentrations tested. Several targeted electrophiles do not lead to the same gross cellular injury CP phenotype. Note the tested electrophiles demonstrate gross cell injury at micromolar, but not nanomolar, compound concentrations. Image scales: 50 μm. Data are mean ± SD of four intra-run technical replicates each performed on separate microplates. Source data are provided as a Source Data file.

KAT inhibitors (hKATIs) are enriched for nonspecific electrophilicity, aggregation, suboptimal potency and selectivity, and cytotoxicity[31]. However, highly potent and specific "next-generation" KAT inhibitors (ngKATIs) have now been reported, including the KAT3 inhibitor A-485 (**468**) and its negative control analog A-486 (**469**), and the KAT6 inhibitors WM-8014 (**470**) and WM-1119 (**471**), and their negative control analog WM-2474 (**472**)[35–37]. Given our previous experiences characterizing hKATIs for assay interference and their association with nonspecific activity, we sought to determine if differences in probe quality could be distinguished by profiling the ngKATIs **468–472** and hKATIs **473–501** using CP and cellular injury reference compounds.

We first profiled the ngKATIs **468–472** for the cell-free liabilities characteristic of many hKATIs to determine whether ngKATIs might be analyzed distinctly from hKATIs. In contrast to hKATIs, **468–472** showed acceptable profiles for potency, selectivity, reproducibility, colloidal aggregation, redox cycling, nonspecific thiol reactivity, chemical instability, light absorption, fluorescence, and quenching (Supplementary Figs. 4, 5; Supplementary Notes). Furthermore, **468–472** were nontoxic and decreased cellular H3K27ac levels as expected in MCF7 and HEK293T cells (Supplementary Fig. 4). By contrast, the interference compounds rottlerin (**478**) and plumbagin (**486**) also decreased cellular H3K27ac levels, but in addition reduced total cellular KAT3B (p300) levels. These data confirm the reported on-target activities of **468–472** and indicate that they are unlikely to exhibit

common assay interference modes. The clean interference profiles of **468–472** support grouping ngKATIs as distinct from hKATIs.

The two KAT inhibitor categories produced distinct CP morphologies. Whereas the ngKATIs **468–472** were CP-inactive and had no effect on cell number, many hKATIs were active and strongly reduced cell numbers in CP (Fig. 3a). More specifically, most hKATIs resulted in significantly increased activity (19/26, 73%) and decreased cell numbers (14/26, 54%) at 20 μM. The ngKATIs occupied different PCA feature-space from most hKATIs, with the latter occupying cluster 9 (cell injury) when tested at higher concentrations that coincide with their reported cellular KAT inhibition activities (Fig. 3a). The summary morphological fingerprints were essentially null for ngKATIs while the hKATIs mirrored cluster 9 (Fig. 3b). Many hKATIs (6/26, 23%) produced the gross injury phenotype (cluster 9) at 20 μM, and even more so at 80 μM (10/17, 59%); a notable subset of hKATIs (9/26, 35%) produced the genotoxic phenotype (cluster 6) at 20 μM.

Inspection of individual KAT inhibitor profiles agreed with these overall observations, with the NSEs L002 (**481**) and NU-9056 (**490**) producing gross injury phenotypes at concentrations similar to other NSEs (Fig. 3c). The prototypical aggregators anacardic acid (**487**) and MG149 (**493**) showed abrupt and concomitant increases in CP activity and decreased cell numbers near their approximate critical aggregation concentrations (CACs). Again, there was a general trend of increased CP activity scores and lower relative cell counts with

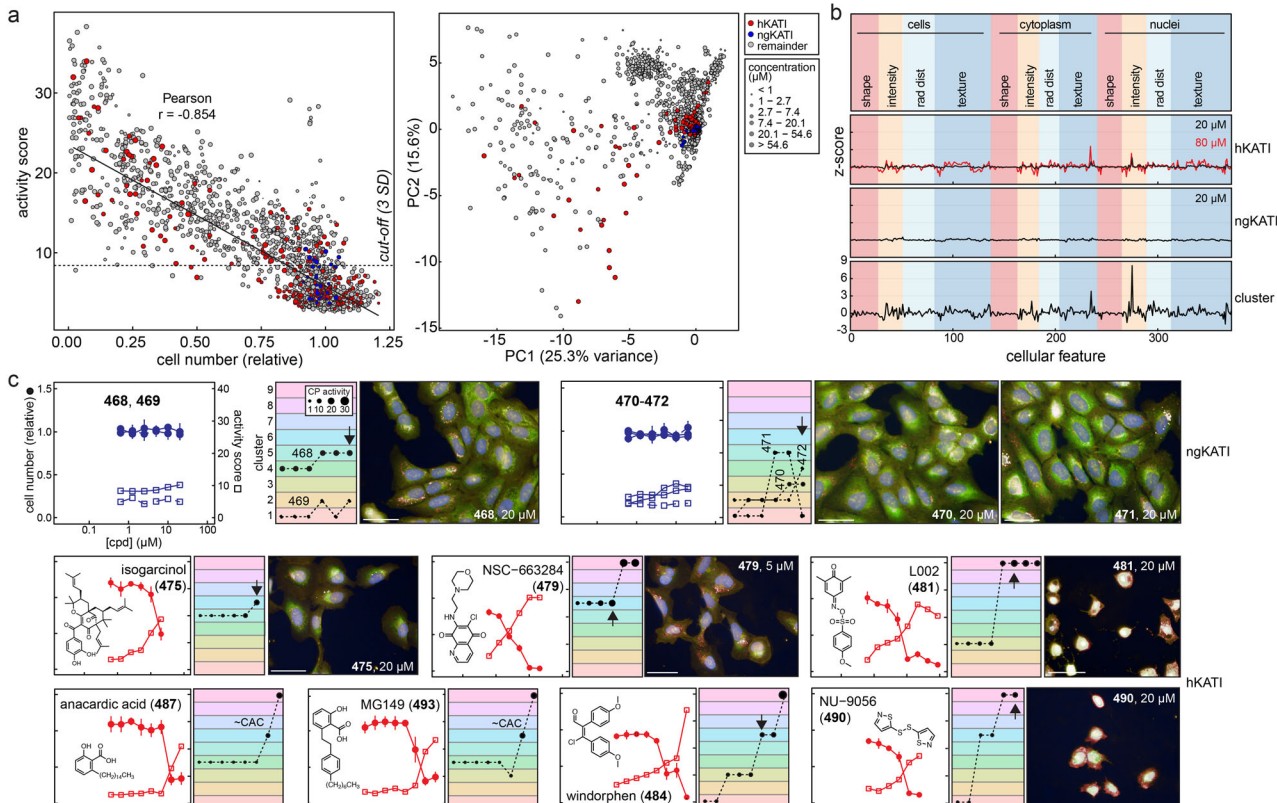

**Fig. 3 | Historical but not next-generation KAT inhibitors produce gross injury phenotypes in cell painting.** A collection of KATIs were each profiled by cell painting (CP) after 24 h of compound exposure in U-2 OS cells. Compared to ngKATIs, many hKATIs are associated with assay interferences, suboptimal specificity, and cytotoxicity. **a** Most hKATIs but not ngKATIs perturb cell number and are scored as bioactive in CP and occupy different feature-spaces by PCA. The cut-off is 3 SD from the mean of DMSO-treated wells using the CP activity (Mahalanobis distances). **b** Reduced CP feature summaries for hKATIs and ngKATIs. Several hKATIs can produce CP phenotypes similar to the gross cell injury phenotype

(cluster 9), especially at higher compound concentrations. The ngKATIs do not cause as pronounced CP phenotypes as hKATIs. **c** Select CP profiles of KAT inhibitors. The ngKATIs **468**−**472** do not cause the gross cellular injury CP phenotype, whereas many hKATIs produce the cell injury CP phenotype at higher compound concentrations. Note the pronounced inverse relationship between relative cell number and CP activity. Image scales: 50 µm. Data are mean ± SD of four intra-run technical replicates each performed on separate microplates. h/ngKATI; historical/ next-generation lysine acetyltransferase inhibitor; PC, principal component. Source data are provided as a Source Data file.

increasing compound concentrations. Notably, many hKATIs were profiled at higher compound concentrations based on their reported cellular potencies.

Interestingly, histone acetylation status and CP morphologies were not correlated based on profiling cellular acetylated histones and KAT3B levels under conditions mimicking the CP assay (Supplementary Fig. 6). These observations are consistent with previous reports that in certain prostate and hematologic cancer cell lines, cellular histone acetylation levels do not necessarily correlate with antiproliferative effects[35]. These data illustrate the subtle but important point that quality, on-target probes do not necessarily produce detectable CP phenotypes, whereas nonspecific compounds can generate significant CP phenotypes. Robust phenotypes for processes like chromatin remodeling may require longer compound treatment times not amenable to standard CP conditions. Overall, the CP and interference profiling data support the use of ngKATIs **468**−**472** as quality epigenetic probes. Other recently reported ngKATIs likely behave similarly. For example, the KAT7 inhibitor WM-3835 contains the same acylsulfonohydrazide scaffold as **470**−**472**[38]. Neither WM-3835 or CPI-1612 (a KAT3 inhibitor) contain red-flag interference chemotypes found in many hKATIs (e.g., quinones, polyphenols)[31,39].

### Cellular health and cell-painting compound profiles

We next sought to characterize the connection between CP-detected cellular injury-based phenotypes and more specific cellular health readouts by multiplexed live-cell imaging under CP-like conditions.

The CP activities and relative cell numbers of 254 profiled compounds (218 cellular injury compounds plus KATIs and electrophiles, based on sample availability and assay throughput) were correlated with culture confluence (phase contrast), caspase-3/7 activation (GFP channel, fluorogenic caspase 3/7 substrate), and cell viability (RFP channel, CytoTox dye, which marks compromised membrane integrity) by live-cell imaging (Fig. 4a). Compounds with the most pronounced changes in cell confluence, caspase 3/7 activation, and compromised membrane integrities were found in PCA clusters 7−9 (Fig. 4a, b). Similar patterns occurred when analyzed by compound category, with hKATIs, MLI-HC, and NSEs exhibiting cellular damage profiles in live-cell imaging, whereas their respective ngKATI, MLI-NC, and NSE-IA counterparts were largely inert (Fig. 4b). Individual compound profiles agreed with these overall observations, where adverse changes in cellular health biomarkers generally increased in magnitude and decreased in time-to-onset with higher compound concentrations (Fig. 4c). These data confirm the cell injury properties of the initial compound profiling collection, and underscore a clear association between strong CP activity scores (phenotypes) and cellular injury.

Live-cell imaging does not include washing steps prior to image acquisition and can potentially enrich for compounds acting by technology-related interferences like compound auto-fluorescence. Indeed, reagent-free counter-screens identified several auto-fluorescent compounds that were excluded from analyses (Supplementary Fig. 7). These observations reinforce the need to carefully inspect live-cell imaging images and time-course data for compound

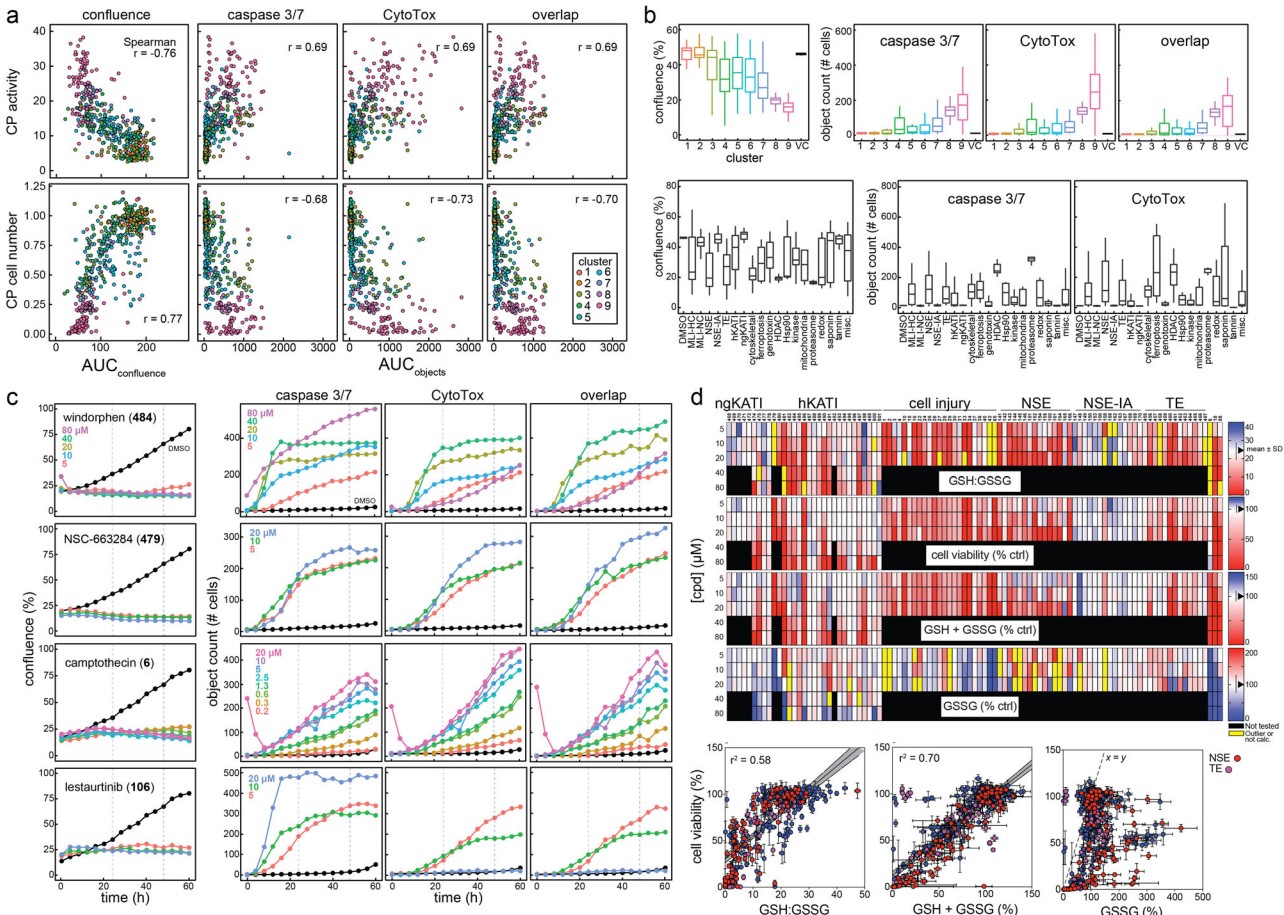

**Fig. 4 | Cellular health and cell-painting phenotypes are correlated.** Compounds tested in cell painting (CP) were profiled for cellular health biomarkers under CP-like conditions to better understand the relationship between CP phenotypes and cellular health. **a** CP activities and relative cell numbers correlate with cellular confluence, caspase 3/7 activation, cell viability, and overlap (cells with caspase 3/7 activation and loss of membrane integrity) after 24-h compound treatment as measured by live-cell imaging. AUCs were calculated from the live-cell imaging assay concentration-response curves for each compound. **b** Breakdown of live-cell imaging cellular health profiles (AUC) by CP phenotypic clusters and cell injury compound groups. The CP phenotypic clusters associated with prototypical cell injury compounds have decreased cellular confluence and increased caspase 3/7 activation and loss of membrane integrity. The cellular health biomarkers vary depending on compound group. Object count indicates the number of cells ("objects") above the signal threshold for each biomarker. The top/bottom of the boxes refer to the first/third quartiles; the whiskers refer to either the most extreme value or 1.5× inter-quartile range from the first/third quartiles, whichever has a smaller absolute value. **c** Select live-cell imaging profiles for cellular health. Cell injury compounds predictably decrease confluence, and can lead to increases in caspase 3/7 activation and membrane disruption within 24 h of compound treatment. Data are mean ± SD of three intra-run technical replicates each performed on separate microplates. **d** Cell injury compounds, most notably electrophiles, decrease the intracellular glutathione (GSH):oxidized glutathione (GSSG) ratio. Black, not tested; yellow, outlier (GSSG > 200%) or not calculated (GSH:GSSG, calculated GSH < 0%). Data are mean (±SD) of five intra-run technical replicates each performed on separate microplates. h/ngKATI; historical/next-generation lysine acetyltransferase inhibitor; NSE-(IA), nonspecific electrophiles (inactive analog); TE, targeted electrophile. Source data are provided as a Source Data file.

interferences, and perform appropriate control experiments. This is important in live-cell imaging when testing a large number of compounds or nonoptimized compound collections that may be enriched for properties linked to assay interferences (poorly soluble or highly aromatic compounds).

Given the importance of electrophilic and oxidative stress on cellular health, we profiled 99 select compounds at several concentrations to characterize the relationships between intracellular glutathione levels under CP-like conditions. The ngKATIs did not grossly perturb glutathione homeostasis, whereas many hKATIs (10/26, 38%), TEs (10/13, 77%), NSEs (12/15, 80%), and cellular injury compounds (14/26, 54%) lowered the GSH:GSSG ratio at 20 μM where most of these compounds were also active in CP (Fig. 4d). Together, these data further demonstrate that CP activity is susceptible to various mechanisms of compound-mediated cellular injury and can be used to refine our understanding of undesirable compounds.

## Time-dependence of cellular injury morphologies

While most analyses focused on 24-h treatment times, we also profiled a subset of compounds after 48-h incubation to determine the effect of compound treatment times on CP readouts. There was strong correlation between replicate compound treatments at each of 24- and 48-h, and high correlation between the pair-wise 24- and 48-h compound treatments (Fig. 5a). There was also a strong correlation between compound treatments with strong CP signals (i.e., CP activity/Mahalanobis distance >10), which could be attributed to the higher signal-to-noise of their CP profiles (Fig. 5b). The individual CP clusters deriving from 24- and 48-h compound treatments were also correlated (Fig. 5c). Comparing the unsupervised clustering of compounds at 24 and 48 h evinced general agreement (entanglement = 0.76; Fig. 5d). Furthermore, cellular health biomarkers from the live-cell imaging experiments were grossly correlated at 24- and 48-h treatment times (Fig. 5e). Inspection of individual compound profiles supported these trends, with most compounds exhibiting similar cell numbers and

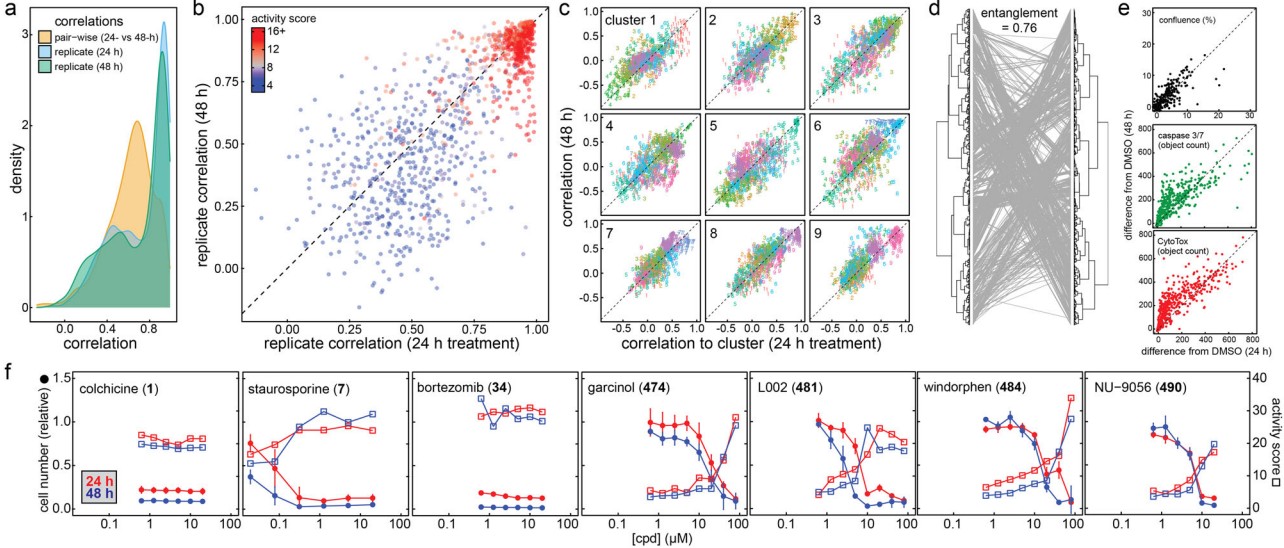

**Fig. 5 | Additional analysis of cell injury compounds in cell painting.** Cellular injury compounds were tested at 24- and 48-h to determine the effect of compound treatment times on cell-painting (CP) readouts. **a** CP phenotypes after 24- and 48-h compound treatments are correlated. **b** Replicates of active CP compounds are correlated at 24- and 48-h compound treatment times. The replicate correlation is defined by the average correlation between each replicate pair (6 comparisons total). There are generally strong correlations between compound treatments with strong signals (red). **c** CP clusters from 24- and 48-h compound treatments are correlated. The horizontal- and vertical-axis correspond to the correlation of the 9 clusters to the 24- and 48-h treatment profiles, respectively. **d** CP profiles from 24- and 48-h compound treatments are correlated. Tanglegram shows connections between compound treatments from 24- and 48-h treatment dendrograms. **e** Cellular health biomarkers (AUC) are grossly correlated at 24- and 48-h treatment times. Compounds that decrease confluence, active caspase 3/7, and decrease membrane integrity at 24 h generally produce similar changes at 48 h. N.B., while many points lie along the parity line, some have larger values at 48 h. **f** Select CP profiles comparing 24- and 48-h compound treatments. Note the trends of relative cell number and CP activity scores are similar at both time points. Data are mean ± SD of four intra-run technical replicates each performed on separate microplates. Source data are provided as a Source Data file.

bioactivities at 24- and 48-h treatment times (Fig. 5f). These data suggest that many grossly cytotoxic compounds can produce detectable CP morphological changes by 24 h of treatment, and that in many cases 24 h is a sufficient compound incubation time to detect gross cellular injury in the standard CP protocol with U-2 OS cells.

### Proposed cellular nuisance control compounds

While it may be tempting to discard compounds that result in low cell numbers, one may unnecessarily eliminate potentially useful bioactive compounds from consideration (e.g., novel and potent anti-neoplastics). In cellular assays, the causative factors for phenotypes are not known a priori and require follow-up experiments[3,40]. Focused chemical libraries composed of bioactive reference compounds can be highly useful for characterizing cellular readouts[41]. Interference characterization is standard in clinical assay validation, and some industrial screening centers even utilize "informer sets" composed of nuisance and cytotoxic chemicals[42,43].

Based on our data and cumulative experience with HTS, we propose an informer set of control compounds to model cell injury phenotypes in HCS and other phenotypic assays including mechanism-based and nonspecific modes of gross cellular injury (Fig. 6; Supplementary Data 1, column "Proposed informer set"). This set would ideally include multiple concentrations of each compound (i.e., nM to μM), focused active and inactive chemical analogs (if available), and multiple chemotypes for each cell injury cluster (especially for cluster 9). Such focused redundancy can mitigate compound-specific effects and assess experimental imprecision[44,45]. Such a proposed set could be adapted to a single 384-well microplate in qHTS format and should only represent an incremental cost in the context of larger-scale campaigns, especially if repeatedly used. This set would ideally be used in parallel with an informer set composed of FDA-approved drugs and high-quality chemical probes to assess for overlapping phenotypes[41].

## Discussion

Here we provide a set of nuisance and cytotoxic compounds of various mechanisms, a dataset documenting their morphological impacts, and a strategy for using them as landmarks alongside novel compounds of interest, or even for assessing an entire compound library. There is a growing interest in phenotypic assays for drug discovery and chemical biology due to their purported potential for improved in vivo and clinical translation[40,46]. Phenotypic and high-content assays such as CP are attractive, target-agnostic approaches for biological annotation of compounds. However, active compounds in cellular assays can act by on- or off-target effects, meaning that without detailed follow-up experiments, readouts are essentially "black boxes"[47]. As a result, a significant obstacle in cellular assays are bioactive compounds that act via undesirable MoAs like cellular injury. Since complex phenotypes are difficult to predict a priori, one practical solution is to include known reference compounds.

To test this solution, we profiled 218 prototypical cytotoxic and nuisance compounds in qHTS format, using CP and companion cellular health assays, to characterize the relationships more systematically between morphology and cellular injury. Targeted and nonspecific electrophiles, along with historical and next-generation KAT inhibitors, served as important case studies. Several important trends emerged: (1) there is a clear relationship between many types of cellular injury and CP activity, (2) nonspecific and suboptimal probes such as hKATIs can produce profound CP phenotypes, (3) compound-mediated cellular damage (e.g., tubulin poisons, genotoxins) can produce robust CP phenotypes, and (4) compound concentration is a key modifier of cellular injury phenotypes such as NSEs, aggregators, and surfactants. The diversity of compounds in cluster 9 (often at high concentrations) may be partially explained by the proposed "cytotoxicity burst" phenomenon whereby compounds are thought to activate multiple stress responses rather than a singular molecular target[48,49]. The clear association between CP activity and cellular

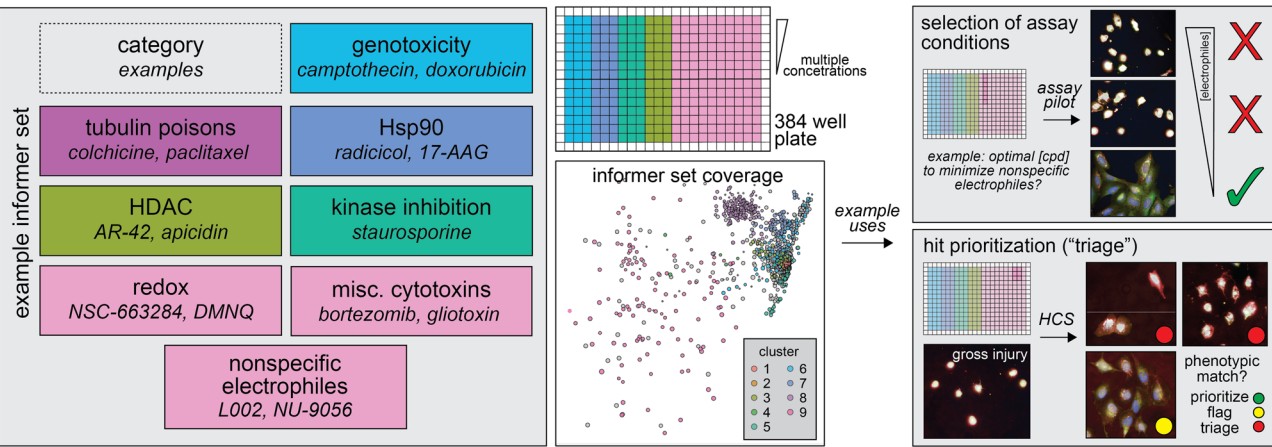

**Fig. 6 | Proposed nuisance compound informer set for use in HCS assays.** Left: an informer set can be assembled using representative compounds from key cellular injury and nuisance mechanisms of action (MoAs). Based on profiling experiments, cellular injury compounds should be tested at several concentrations. The large number of cell-painting (CP) phenotypes associated with gross cell injury (cluster 9, note relative area compared to other clusters) suggests users should include a set of compounds that represent multiple undesirable MoAs, along with several compounds per undesirable MoA. Right: When an informer set is used during the assay optimization phase, more optimal experimental conditions can be selected to reduce the incidence of unwanted MoAs such as nonspecific electrophiles. When an informer set is used compound prioritization phase ("hit picking", "HTS triage"), compounds sharing phenotypes with unwanted MoAs can be prioritized according to the desired compound characteristics. See also Supplementary Data 1 for a proposed informer set for CP and potentially other cellular and high-content assays. HCS, high-content screen; HDAC, histone deacetylase.

damage suggests active compounds should be subjected to cellular health profiling.

An important observation from testing the historical and next-generation KAT inhibitors is that some high-quality bioactive compounds do not lead to active CP profiles at concentrations at which they modulate their target. This suggests there are limits to the mechanistic space that can be captured by CP. In some cases, bioactive compounds may simply not produce detectible morphological changes in cells, whether at all or within the usual 24- to 48-h compound treatment windows of the conventional CP assay. Therefore, it should not be assumed that a bioactive compound will produce a CP phenotype. More sophisticated image analyses and/or alternative experimental protocols may improve the number and type of phenotypes that can be reliably detected with CP and similar morphological assays. The large number of phenotypes caused by cellular injury compounds (cluster 9) suggests a significant portion of CP morphologies may be affected by compound-mediated cellular injury. For any given morphological assay, this could be estimated by testing a diverse selection of bioactive probes and cellular injury control compounds.

Future efforts could focus on the association between specific chemotypes and CP profiles, as well as phenotypic profiles in general. In this work, potent electrophiles (quinones, benzothiophene 1,1-dioxides, unstable succinimides, maleimides, etc.) produced strong CP profiles associated with cellular injury, whereas relatively weaker electrophiles (acrylamides) occasionally produced similar profiles at higher micromolar concentrations. Given our previous experiences determining structure-interference relationships with problematic chemotypes in biochemical assays[2], the generalization of chemotypes with specific CP profiles would benefit from testing a variety of analogs with and without the suspected problematic structural feature.

This resource expands upon existing open-source CP datasets due to its concentration-response format, time-course data, accompanying cellular health profiling data, and the intentional profiling of prototypical cellular injury compounds. The large volumes of data generated by high-content assays is a practical barrier that hinders reproducibility and scientific exchange in the field[50]. To address this barrier, the entire 11 TB of raw images from this study (including all solvent controls), representing 1.04 million images in total, and the associated metadata are available via the open-source Image Data Resource [https://idr.openmicroscopy.org][51]. This resource should enhance compound prioritization in complex cellular assays. Potential uses for this dataset include HCS and image-processing method development, MoA studies, informer set design, and compound triage[52,53]. These data can be re-analyzed by alternative methods such as focusing on individual features or more complex analyses such as the point-of-departure metric[18,54].

Compound-induced phenotypes may be cell line-dependent phenotypes, and may also depend on other experimental factors including treatment time, compound concentration, and culture media composition[55,56]. Although we only profiled one cell line, this approach is likely generalizable to other biological systems and profiling assays. Supporting evidence includes: (1) the similar behaviors of hKATIs and prototypical interference compounds in MCF7, HEK293, and U-2 OS cells; (2) the profiled compounds were chosen based on activities unrelated to U-2 OS cells; and (3) the biological targets of these cellular injury compounds tend to be common amongst human cell lines. Our findings are complementary to but consistent with previous reports, including the robust phenotypes of cytoskeletal poisons, genotoxins, and other classes of cellular injury compounds[13,45,55-58]. Furthermore, reference compounds can show similar phenotypes across different cell lines in HCS assays[13,18,59,60]. Notably, our findings also mirror a profiling effort that combined gene expression and cell death profiles, which found compound clusters consisting of microtubule modulators, electrophiles, and genotoxins[61].

Lastly, we propose a framework (Supplementary Data 1, column "Proposed informer set") for constructing cellular injury informer sets applicable beyond CP, from alternative high-content morphology assays to orthogonal cellular assay technologies like gene-expression profiling and metabolomics. This proposed set should serve as a useful starting point for practitioners, and could be subject to future improvements as additional evidence is generated by the scientific community. While practitioners could build their own custom sets, using a common set (or even subset) of reference cellular injury compounds may benefit the scientific community as a whole and enable data harmonization. It is likely that modifications to the set may be needed for specific assays and model systems, such as the addition of compounds with other cytotoxic MoAs, or the removal of

redundant compounds if throughput or cost is a concern. This set could have several applications in cellular assays. During the assay development and optimization phase, such a set could guide the choice of experimental conditions to minimize the selection of unwanted MoAs such as NSEs. During the screening phase, such a set could guide compound triage and follow-up experiments, especially if new compounds share phenotypes with unwanted MoAs. Used in parallel with screening, such a set could facilitate real-time compound evaluation of high-risk compounds, and potentially reduce the necessity of certain follow-up counter-screens.

In terms of practical use, compounds that produce similar phenotypes as cellular injury compounds can be prioritized (i.e., if assaying for novel or mechanism-based cytotoxins) or triaged (i.e., de-prioritizing tubulin or mitochondrial poisons). In one study, compounds profiled with the L1000 transcriptome profiling assay and CP, cytotoxic compounds produced robust signatures in both techniques[45]. This further suggests that the proposed approach can be applied to other assays and cell types. Nuisance and cellular injury compounds will likely show some assay-dependent effects, as nuisance compound behavior can vary between targets and technologies even with comparatively simpler biochemical assays. This further supports the value of repeatedly using an informer set. Given the complexities of cellular nuisance compounds and their context dependence, it is difficult at this point to quantify the sensitivity and accuracy of such an informer set in predicting whether an active compound is acting by a nuisance mechanism. The use of such a standardized set by the chemical biology and drug discovery communities should help to address this important question.

Future iterations focused on cellular injury could include MoAs not profiled, such as chelation, metal poisoning, membrane bilayer disruptors, and lysosomotropic agents. Future work could also characterize the effects of compound technology-related interferences ("artifacts") such as auto-fluorescent or quenching compounds[62,63]. We envision that collaboration between academic, industry, and government groups performing phenotypic screening can enhance our proposed informer set by nominating additional compounds and performing additional validation in orthogonal phenotyping assays and biological systems[64].

## Methods

### Compounds and reagents

Sources of KAT inhibitors are listed (Supplementary Table 1). Additional compounds were obtained from commercial vendors and the Broad Institute chemical screening library. Test compounds were typically prepared as 10 mM stock solutions dissolved in neat dimethyl sulfoxide (DMSO) and stored under vacuum seals in either −20 °C or a light-shielded desiccation chamber at RT. All were subjected to internal quality control[65]; most demonstrated >90% purity and detection of an expected parent ion by ultra performance liquid chromatography−tandem mass spectrometer (UPLC-MS) (Supplementary Data 1).

### Cell lines

HEK293T cells were gifted from Dr. Sam Benchimol (York University; ATCC, cat # CRL-3216); MCF7 and U-2 OS cells were obtained directly from ATCC (cat # HTB-22 and HTB-96, respectively). HEK293T and MCF7 cells were used to remain consistent with our previous report[31]. U-2 OS cells were used for CP because they form monolayers highly amenable to single-plane high-content imaging and have been profiled extensively at several institutions[7,14]. U-2 OS cells do not contain significant genetic perturbations in KAT3 (Supplementary Notes). Cell-line identities were confirmed by short tandem repeat profiling (ATCC Cell Line Authentication Service or provided by vendor) upon receipt, and *Mycoplasma* contamination was assessed monthly with the MycoAlert PLUS *Mycoplasma* Detection Kit (Lonza, cat # LT07-701).

### Cell painting

Cell painting (CP) was adapted from previous reports[6,7]. New U-2 OS cells were purchased directly from ATCC for each CP experiment and used within the first ten passage numbers. U-2 OS cells were cultured in DMEM (high glucose, GlutaMAX, HEPES; Thermo Fisher, cat # 10564011) supplemented with 10% fetal bovine serum (FBS, v/v; Gibco, cat # 26140079), penicillin (100 U mL$^{-1}$), and streptomycin (100 µg mL$^{-1}$), and maintained in a 37 °C, 5% CO$_2$ humidified incubator. Cells were dispensed by Thermo Multidrop into 384-well clear-bottom imaging plates (CellCarrier-384 Ultra; PerkinElmer, cat # 6057300). Each well contained approximately 750 (48 h treatment) or 1,500 (24 h treatment) cells in 50 µL complete media. Cell counts were determined by Countess II automated cell counter (Thermo Fisher) using 0.4% trypan blue solution (Thermo Fisher, cat # T10282). Seeded microplates were incubated for 24 h at 37 °C, then treated with compounds or vehicle controls dispensed by pin tool transfer via CyBi-Well robot. Compounds were typically tested at six concentrations with 2-fold serial dilutions, ranging from 20 µM to 625 nM final compound concentrations. Each plate contained four positive control compounds (**1**–**4**; colchicine, nocodazole, radicicol, wortmannin) at six concentrations each and 116 vehicle control wells (Supplementary Notes). Following the addition of compounds, cells were incubated at 37 °C for either 24 or 48 h.

A 1 mM DMSO solution of MitoTracker Deep Red FM (Invitrogen, cat # M22426) was added to pre-warmed complete media to make staining solution 1 (SS1) with final concentrations 500 nM MitoTracker. A 1 mg mL$^{-1}$ solution in 0.1 M aqueous sodium bicarbonate of concanavalin A-Alexa Fluor 488 conjugate (Invitrogen, cat # C11252), a 200 U mL$^{-1}$ methanol solution of phalloidin-Alexa Fluor 568 conjugate (Invitrogen, cat # A12380), a 1 mg mL$^{-1}$ dH$_2$O solution of wheat germ agglutinin (WGA)-Alexa Fluor 555 conjugate (Invitrogen, cat # W32464), a 16.2 mM aqueous solution of Hoechst 33342 (Invitrogen, cat # H3570), and a 5 mM DMSO solution of SYTO 14 green fluorescent nucleic acid stain (Invitrogen, cat # S7576) were combined in 1X HBSS (prepared from 10X solution, Thermo Fisher, cat # 14065-056; filtered) supplemented with 1% bovine serum albumin (BSA; m/v) to make staining solution 2 (SS2) with final concentrations 100 µg mL$^{-1}$ concanavalin A, 0.5 U mL$^{-1}$ phalloidin, 60 µg mL$^{-1}$ WGA, 8.1 µM Hoechst, and 3 µM SYTO 14.

Compound-treated cells were prepared for fixation and staining by first removing 40 µL of media from each microplate well using a BioTek ELK405 automated plate washer. To each well, 30 µL of SS1 were dispensed by Multidrop, followed by incubation for 30 min at 37 °C. Cells were then fixed by dispensing 10 µL per well of 16% aqueous paraformaldehyde via Multidrop, followed by incubation for 20 min at RT. Next, wells were washed with 70 µL 1X HBSS. Cells were then permeabilized by adding 30 µL per well of 0.1% Triton X-100 (v/v) in 1X HBSS and incubated for 15 min at RT. Wells were then washed with 70 µL 1x HBSS. Permeabilized cells were then stained by dispensing 30 µL SS2 per well via Multidrop followed by incubation at RT for 30 min. Wells were washed with 70 µL 1x HBSS without a final aspiration, then the plates were manually sealed with adhesive foil for subsequent imaging.

Cells were imaged using an Opera Phenix High-Content Screening System (PerkinElmer) equipped with Harmony software (version 4.9) in wide-field mode with a water-immersion 20x objective and five excitation/emission laser wavelengths: 405/435-480 (Hoechst), 488/500-550 (concanavalin A), 488/570-630 (SYTO 14), 561/570-630 (phalloidin and WGA), and 640/650-760 (Mito-Tracker) nm. Photobleaching of low-intensity dyes was mitigated by imaging in the following channel order: MitoTracker, WGA, phalloidin, SYTO 14, concanavalin A, and Hoechst 33342. Nine sites were imaged per well in a 3 × 3 array with laser-based autofocus on the first site per well. Concentration-response data are mean ± SD from four intra-run technical replicates each performed on separate microplates.

## Cell-painting analyses

Morphological features were extracted from the raw CP images using a freely available CellProfiler software pipeline provided by the Broad Institute Imaging Platform[7,66]. Images were corrected for uneven illumination, then CellProfiler (version 2.1.1) was used to locate and segment cells into nuclei and cytoplasm, after which the size, shape, texture, intensity, local density, and radial distributions were measured for nuclei, cytoplasm, and entire cells. To obtain profiles for each compound, the morphological features of compound-treated cells were calculated for each well field, then averaged per well, and then normalized by calculating robust $z$-score-like values based on the population of individual DMSO-treated cells found on the same plate.

To determine the compound activities ("activity scores"), Mahalanobis distances of each compound profile were calculated from vehicle-treated well profiles[67]. The profiles for all replicates of a compound were first combined with the corresponding DMSO-control wells into a matrix of dimensions $m \times n$, where the rows $m$ represent the individual wells and the columns $n$ represent the profiling features. PCA was performed on this matrix to obtain a new matrix $P$ with the principal components as the columns. For each of these matrices, the first $q$ principal components were taken that could explain ≥90% of the variance. This matrix $P$ was separated into treatment and control matrices and for each part a covariance matrix was calculated. Each of the two covariance matrices (treatment and control) was weighted by the number of samples in each matrix, and the sum of the resulting matrices was used to calculate the Mahalanobis distance. The R *cytominer* package was then used to reduce the number of redundant features by removing those which were highly correlated. After processing, 372 non-redundant cellular features were used in the subsequent analyses.

The principal component analyses were performed using the *R stats* package (version 3.6.1) using non-aggregated treatment profiles (i.e., compounds, concentrations, and replicates separate) with scaling and without centering. These principal components were used for all PCA plots; for PCA plots containing only a subset of the data or using aggregated data, points were transformed into this space using the loadings of these principal components. Unsupervised hierarchical clustering was performed using the *R stats* package (version 3.6.1), using the values [1−Pearson correlation] as the pairwise distances between each treatment condition and Ward's method (minimal increase of sum-of-squares). Entanglement was analyzed by comparing hierarchical clustering of the 24- and 48-h compound treatment duration CP data by Ward's method using the *R dendextend* package (version 1.13.2). To generate "dot plots," the values for the activity scores were averaged across all categories falling under a particular cluster (obtained from hierarchical clustering). To calculate the similarity of historical MLI CP data to the observed cellular injury phenotypes obtained from hierarchical clustering, only features belonging to the reduced feature set and shared between the historical data and this study were considered. Phenotypes are calculated as the average signature of each cluster, where the average value for each feature across all cluster members are taken, and the vector comprising these average values is used as the cluster phenotype (see also Supplementary Fig. 2). The number of clusters was based manually on previous experiences to optimize the quality of the tubulin compound cluster, historically the most robust phenotype in previous experiments.

## Live-cell imaging

U-2 OS cells were cultured similarly to the CP protocol but were modified for seeding density (750 cells in 25 μL media per well), microplate (384-well, tissue culture-treated, black polystyrene, clear flat-bottom microplates; Corning, cat # 3764BC), and media (F-12K, ATCC, cat # 30-2004; supplemented with 10% FBS, 100 U mL$^{-1}$ penicillin, and 100 μg mL$^{-1}$ streptomycin). This media was selected because it reduced background fluorescence in the GFP channel due to

riboflavin[68]. Seeded microplates were then incubated for 24 h at 37 °C, followed by the addition of sterile-filtered 25 μL media to each well containing live-cell imaging reagents (Incucyte Cytotox Red Reagent, final concentration 250 nM, cat # 4632; Incucyte Caspase 3/7 Green Reagent, final concentration 5 μM, cat # 4440). A reagent-free technology interference counter-screen was performed similarly, substituting the reagents with identical solvent control. Cells were then immediately treated with 100 nL of compounds or vehicle controls dispensed by pin tool transfer via CyBi-Well Vario with a constant DMSO concentration of 0.2% (v/v). Most compounds were tested at three concentrations (20, 10, and 5 μM final concentrations). Each microplate contained reagent-free control wells to correct for cellular auto-fluorescence. Following compound addition, cells were incubated at 37 °C for 60 h and imaged every 4 h with an Incucyte S3 Live-Cell Analysis System (Essen Biosciences) utilizing 10x objective and 300 and 400 ms GFP and RFP channel acquisition times, respectively. Live-cell images were processed in Incucyte Analysis Software (Essen Biosciences) using top-hat background correction. Images were analyzed for confluence, total area of green/red/green+red (overlap) fluorescence, number of green/red/green+red (overlap)-positive objects, and integrated green/red fluorescence. Data are mean ± SD from three intra-run technical replicates each performed on separate microplates. To analyze compounds tested in concentration-response format, area under the curve (AUC) was calculated[69].

## Intracellular glutathione quantification

U-2 OS cells were cultured similarly to the aforementioned CP protocol, except for seeding density (750 cells in 25 μL media per well) and microplate (384-well, tissue culture-treated, low-volume, white polystyrene, flat-bottom microplates; Corning, cat # 8867BC). Seeded microplates were then incubated for 24 h at 37 °C, then treated with 100 nL compounds or vehicle controls dispensed by pin tool transfer via CyBi-Well Vario with a constant DMSO concentration of 0.4% (v/v). Most compounds were tested at three concentrations (20, 10, and 5 μM final concentrations). Following the addition of compounds, cells were incubated at 37 °C for 24 h. Each microplate contained triplicate 10-point GSH standards (6.4 μM to 13 nM by 2-fold serial dilutions) for quantifying GSH. After compound treatment, total intracellular glutathione (reduced, GSH and oxidized, GSSG) concentrations and oxidized glutathione (GSSG) were quantified in parallel on separate microplates with the GSH/GSSG-Glo kit (Promega, cat # V6611) per manufacturer protocol, except that all reagents were diluted threefold in PBS. Cell viability was quantified in parallel on separate microplates by Cell Titer Glo (Promega, cat # G7570) per manufacturer protocol. Luminescence was measured on an Envision 2105 plate reader (PerkinElmer) with 400 ms acquisition times. Data were corrected for background luminescence using cell-free control wells and for row/column effects by uniformity plates. Data are from five intra-run technical replicates each performed on separate microplates.

## Biochemical KAT selectivity

ngKATIs were tested for biochemical selectivity versus six KATs. Assay conditions were as follows: KAT2A (hGCN5, 497-662 aa), 1 nM enzyme, 10 μM biotin-H3 1–25, 2.5 μM [$^3$H]-acetyl-CoA; KAT3B (P300, 1195-1662 aa), 2 nM enzyme, 5 μM biotin-H3 1–25, 2.5 μM [$^3$H]-acetyl-CoA; KAT1 (HAT1, 20-341 aa), 1 nM enzyme, 1 μM biotin-H4 1–24, 5 μM [$^3$H]-acetyl-CoA; KAT6A (MOZ/MYST3, 472-793 aa), 10 nM enzyme, 10 μM biotin-H4 1–24, 5 μM [$^3$H]-acetyl-CoA; KAT5 (TIP60, 1-513 aa), 10 nM enzyme, 1 μM biotin-H4 1–24, 0.25 μM [$^3$H]-acetyl-CoA; KAT8 (HMOF/MYST1, 2-458 aa), 20 nM enzyme, 10 μM biotin-H4 1–24, 1 μM [$^3$H]-acetyl-CoA. For KAT3B testing, buffer conditions were 100 mM HEPES, pH 8.0, 2 mM DTT, 100 mM KCl, 80 μM EDTA, 40 μg mL$^{-1}$ BSA (m/v), 0.01% Triton X-100 (v/v). For KAT1/3B/5/6 A/8 testing, buffer conditions were 20 mM tris, pH 8.0, 5 mM DTT, 0.01% Triton X-100 (v/v). Reactions were performed at 23 °C for 20 min. Final DMSO concentration was

constant at 2.0% (v/v). Percent activity represents acetylation relative to vehicle control reactions. Concentration responses were analyzed by nonlinear least-squares regression fits to a four-parameter logistic ("4PL") equation. KAT profiling data are mean of two intra-run technical replicates. Concentration-response KAT6A data are mean ± SD from three intra-run technical replicates.

## Biochemical KAT3B activity

Inhibition of KAT3B acetyltransferase activity by ngKATIs was assessed by a separation-based assay[70]. Reactions consisted of buffer (50 mM HEPES, pH 7.5, 50 mM NaCl, 2 mM EDTA, 2 mM DTT, 0.05% Triton-X-100) with KAT3B (P300, 150 nM) and FITC-Ahx-RGKGGKGLGKGG [Ahx = 6-aminohexanoic acid] substrate (2 µM) were plated in 384-well microplates and equilibrated at RT for 10 min in the presence or absence of inhibitor. Reactions were initiated by addition of acetyl-CoA (1 µM final concentration) with 30 µL final assay volume and quenched during steady-state kinetics (<15% product accumulation) by addition of 5 µL of 0.5 M neutral hydroxylamine. Quenched reaction aliquots were then transferred to a PerkinElmer Lab-Chip EZ-Reader instrument for microfluidic electrophoresis and fluorometric analysis. Optimized separation conditions were downstream voltage −500 V, upstream voltage −2500 V, and pressure −1.5 psi. Percent conversion is calculated by ratiometric measurement of substrate/product peak heights, corrected for nonenzymatic background acetylation. Percent activity KAT3B represents acetylation relative to vehicle control. Concentration responses were analyzed by nonlinear least-squares regression fits to a 4PL equation. Data are mean ± SD from three technical replicates.

## ALARM NMR

ngKATIs were tested by (a La assay to detect reactive molecules by nuclear magnetic resonance) for protein thiol reactivity as previously described[71]. The human La antigen (aa 100−324, T302N) was expressed in *Escherichia coli* Rosetta cells (Novagen) and cultured in M9 minimal media. The La antigen was labeled by adding $^{13}$C-labeled amino acid precursors ([3-$^{13}$C]-α-ketobutyrate and [3,3′-$^{13}$C]-α-ketoisovalerate sodium salts; Cambridge Isotope Laboratories) to culture medium 30 min before IPTG induction. Bacteria were harvested after induction at 25 °C for 8 h, followed by lysis via French press. Labeled La antigen was purified by standard Ni-NTA bead purification. The La antigen product was dialyzed (25 mM sodium phosphate, pH 7.0, 5 mM DTT) in three 16 h cycles at 4 °C with gentle stirring. Aliquots were flash-frozen in liquid N$_2$ and stored at −80 °C until further use. Before use, hLa protein was reduced by incubating with 20 mM DTT at 37 °C for 1 h, then dialyzed (25 mM sodium phosphate buffer, pH 7.0, no DTT) in three 16 h cycles at 4 °C with constant nitrogen bubbling and with gentle stirring. The [$^{1}$H-$^{13}$C]-HMQC spectra were acquired in 25 mM sodium phosphate buffer, pH 7.0, 10% D$_2$O (v/v; CIL) ± 200 µM test compounds ± 20 mM non-deuterated DTT. Final concentration of DMSO was 4.0% (v/v). Reaction solutions were incubated at 37 °C for 1 h and then 30 °C for 15 h before obtaining spectra. Data were recorded at 310 K on a Bruker UltraShield 700 MHz NMR spectrometer equipped with a Bruker 1.7 mm TCI Cryoprobe and Bruker SampleJet autosampler. Samples were tested at 50 µM protein concentrations using 16 scans, 2048 complex points in F2, and 80 points in F1 using standard protein [$^{1}$H-$^{13}$C]-HMQC and water suppression pulse sequences. Sample tubes were inspected for gross compound precipitation. NMR data were analyzed in Bruker TopSpin (version 4.0.7). Reactions were normalized to DMSO controls. Nonreactive compounds were identified by the absence of chemical shifts or changes in peak intensities ($^{13}$C-methyl) ± 20 mM DTT. Reactive compounds induced chemical shifts and decreases in peak intensities in certain diagnostic peaks in the absence of DTT.

## Chemical stability and GSH adducts by UPLC-MS

Gross compound stability of ngKATIs was assessed by incubating parent compound (20 µM final concentrations) in PBS, pH 7.4 at 37 °C for 4 and 24 h. Samples were spiked with fluconazole internal standard (10 µM final concentration, Cerilliant), then diluted with equivolumetric amounts of MeOH to mitigate ion suppression by PBS, then passed through 0.2 µm pore size syringe filters. Samples were also compared to otherwise identical samples containing parent compounds incubated in neat MeOH instead of buffer. Samples were analyzed using a Waters ACQUITY UPLC system using a BEH C18 2.1 × 50 mm column. Samples were injected by an autosampler in 5 µL sample volumes. The flow rate was 0.250 mL min$^{-1}$ with a standard gradient starting at 95% Solution A (950 mL H$_2$O, 50 mL MeCN, 1 mL formic acid) and ending with 100% Solution B (1000 mL MeCN plus 1 mL formic acid) over 2.0 min. The samples were monitored simultaneously by a PDA detector and a ZQ mass spectrometer (electrospray, positive and negative modes). Chromatograms and mass spectra were qualitatively analyzed for the formation of new peaks and/or loss of parent signal relative to internal standard. To detect GSH adducts, test compounds (20 µM final concentrations; 1 eq) and reduced L-glutathione (20 eq) were incubated in PBS, pH 7.4 at 37 °C for 4 h. Samples were prepared identically to the chemical stability studies, but the internal standard was omitted to avoid possible interference with detecting potential compound-GSH adducts. Chromatograms and mass spectra were qualitatively analyzed for characteristic compound-GSH adduct ions by examining the PDA chromatograms for new peaks and the ion chromatograms for loss of the GSH ion (e.g., 307 $m/z$). CPM (Sigma, catalog # C1484) was used as a positive GSH-reactive control compound. UPLC-MS data were analyzed in MestReNova (version 14.1.0-24037).

## Fluorescence intensity thiol reactivity counter-screen

The ngKATIs were tested for non-proteinaceous thiol reactivity using adaptions of previous procedures[2,72]. Thiol-free buffer (25 mM sodium phosphate, pH 7.0, 0.01% Tween-20 v/v) was dispensed in 10 µL volumes into black polypropylene 384-well round-bottom microplates (Agilent, cat # 201290-100) via Multidrop. DMSO, 10 µM N-ethyl maleimide (NEM, Sigma, cat # E1271), and 250 µM BHQ-10 (carboxylic acid, Biosearch) were used as negative, positive reactivity, and positive light interference controls, respectively. Compounds and positive controls (NEM) were transferred to assay plates in 100 nL volumes by pin-tool via CyBi-Well Vario 384/60 (CyBio). Final DMSO concentration was constant at 2.5% (v/v). GSH, CoA, and NAC were freshly prepared as 2 µM solutions in buffer (25 mM sodium phosphate, pH 7.0, 0.01% Tween-20 v/v) and dispensed to the aforementioned microplates in 10 µL volumes via Multidrop (1 µM thiol, final concentration). After thermal sealing (Agilent PlateLoc), microplates were incubated for 90 min at 37 °C in an incubator oven, followed by the addition via Multidrop of 10 µL solution containing 12 µM thiol-reactive probe CPM (Sigma, cat # C1484) prepared in 1:1 DMSO:water. After incubation at RT for 5 min, thiol reactivity was quantified by measuring fluorescence intensity ($\lambda_{ex}$ 405 nm, $\lambda_{em}$ 530 nm) on a SpectraMax M3 plate reader (Molecular Devices; PMT automatic gain, 10 flashes per well). Compounds signals were background-corrected by subtracting the mean negative-control plate control signals. Data are mean ± SD from three intra-run technical replicates performed on the same microplate.

## AmpC aggregation counter-screen

KAT inhibitors were assessed for aggregation using a modified AmpC β-lactamase counter-screen[73]. Recombinant *E. coli* AmpC was obtained from Rosetta cells using a published protocol[74]. The purified protein product was >95% pure by SDS-PAGE analyses and migrated identically to an AmpC standard (Shoichet lab). The enzymatic assay was performed in 50 mM sodium phosphate, pH 7.0 in clear cyclic olefin copolymer 384-well microplates (Aurora, cat # 3030-00330) in 75 µL

reaction volumes. Compounds were tested in triplicate at 4, 11, 33, and 100 μM final concentrations in buffer ± freshly added 0.01% Triton X-100 (v/v). Final concentration of DMSO was 1.0% (v/v). Compounds were incubated with 5 nM AmpC in 73.5 μL reaction buffer for 5 min at RT, followed by the addition of 1.5 μL of nitrocefin substrate (Cayman, cat # 15424) dissolved in DMSO (100 μM initial substrate concentration). Reaction solutions were gently mixed by multichannel pipette. Reaction progress was continuously monitored by absorbance at 482 nm for 5 min at RT on a SpectraMax M3 plate reader, and percent activity was calculated from reaction rates (slope). Percent activity was normalized to DMSO-only controls after background subtraction with an enzyme-free reaction. Avibactam (Cayman, cat # 22825), non-aggregation AmpC-positive control. Statistical significance ($p < 0.05$) was evaluated without assuming consistent SD using two-tailed Student's t-test and the Holm-Sidak method to control for multiple comparisons. Data are mean ± SD from four intra-run technical replicates.

## MDH aggregation counter-screen
KAT inhibitors were also assessed for aggregation using a modified malate dehydrogenase (MDH) counter-screen[75]. The enzymatic assay was performed in 50 mM sodium phosphate, pH 7.0 in clear cyclic olefin copolymer 384-well microplates (Aurora, cat # 3030-00330) in 75 μL reaction volumes. Compounds were tested in triplicate at 100 μM final concentrations in buffer ± freshly added 0.01% Triton X-100 (v/v). Final concentration of DMSO was 1.0% (v/v). Compounds were incubated with 1 nM porcine heart MDH (EMD Millipore, cat # 442610) in 73.5 μL reaction buffer for 5 min at RT, followed by the addition of 1.5 μL of substrate (200 μM oxaloacetate and 200 μM NADH final concentrations; derived from fresh 20 mM stocks in 50 mM sodium phosphate, pH 7.0) and then gentle mixing with multichannel pipette. Reaction progress was continuously monitored by absorbance at 340 nm for 5 min at RT on a SpectraMax M3 plate reader, and percent activity was calculated from reaction rates (slope) and normalized to DMSO-only controls. Statistical significance ($p < 0.05$) was evaluated without assuming consistent standard deviation using two-tailed Student's t-test and the Holm-Sidak method to control for multiple comparisons. Data are mean ± SD from four to eight intra-run technical replicates.

## DLS aggregation counter-screen
Dynamic light scattering was performed as previously described[75]. DMSO stocks of KAT inhibitors were diluted in filtered 50 mM potassium phosphate, pH 7.0, final concentration 1% DMSO (v/v). Light scattering was recorded using a DynaPro Plate Reader II system (Wyatt Technology) with a 60-mW laser at 830 nm, 158° detection angle, and automatically adjusted laser power. Notably this instrument is configured with a larger-width laser beam width optimized for detecting large colloidal particles (BK Shoichet lab, USCF). Data were acquired and processed by DYNAMICS software (Wyatt, version 1.7). Cut-off for colloidal aggregation is $10^6$ counts sec$^{-1}$. Data are mean ± SD from two or three intra-run technical replicates performed on the same microplate.

## Redox activity counter-screen
ngKATIs were assessed for hydrogen peroxide production using a horseradish peroxidase-phenol red counter-screen[76]. Testing was performed in buffer (50 mM tris, pH 7.0) plus 0.01% Triton X-100 (v/v) in clear cyclic olefin copolymer 384-well microplates (Aurora, cat # 3030-00330) in 60 μL reaction volumes. Compounds were tested at 250 μM final concentrations. Final concentration of DMSO was constant at 2.5% (v/v). Compounds were incubated in 40 μL reaction buffer (±1 mM DTT final concentration) for 20 min, followed by the addition of 20 μL solution containing phenol red and horseradish peroxidase (Sigma) dissolved in reaction buffer. Final concentrations of phenol

red and horseradish peroxide were 280 μM and 60 μg mL$^{-1}$, respectively. The reaction solution was allowed to incubate for 20 min at RT, followed by the addition of 10 μL of 1 M sodium hydroxide via multichannel pipette to quench the reaction. After 10 min incubation at RT, hydrogen peroxide was quantified by measuring absorbance at 610 nm on a SpectraMax M3 plate reader. DMSO and 100 μM hydrogen peroxide were used as negative and positive controls, respectively. NSC-663284 (**479**; Cayman, cat # 13303) and 4-amino-1-naphthol HCl (**67**; Oakwood Chemical, cat # 013411) were used as positive compound controls[77,78]. Data are mean ± SD from three intra-run technical replicates performed on the same microplate.

## Light absorbance counter-screen
ngKATIs were assessed for light absorption between 200 and 750 nm at 100 μM final compound concentrations in filtered sodium phosphate buffer (50 mM sodium phosphate, pH 7.0). Final concentration of DMSO were constant at 1.0% (v/v). Compounds were allowed to incubate at RT in buffer for 10 min in UV-transparent half-area 96-well microplates (Corning, cat # 3679). Absorbance spectra were then obtained using a SpectraMax M3 microplate reader at 25 °C using buffer plus DMSO as blank.

## Auto-fluorescence counter-screen
ngKATIs were assessed for auto-fluorescence using an adaption of published procedures[79]. Briefly, fluorophore standards consisted of AlexaFluor 350 (carboxylic acid, Invitrogen, cat # A33076), AlexaFluor 488 (carboxylic acid, Invitrogen, cat # A33077), AlexaFluor 647 (carboxylic acid, Invitrogen, cat # A33084), Texas Red (succinimidyl ester, Invitrogen, cat # T6134), fluorescein (Sigma, cat # F2456), and resorufin (Sigma, cat # 424455). Test compounds were tested in triplicate at six final concentrations (32 nM to 100 μM via five-fold serial dilutions). Fluorophores were tested in triplicate at five to seven final concentrations (0.5 nM to 3 μM). Final concentration of DMSO was constant at 2.0% (v/v). Compounds and fluorophore standards were prepared as serial dilutions from 10 mM DMSO stock solutions, then transferred to 384-well non-binding surface black polystyrene microplates (Corning, cat # 3575) via multichannel pipette. Plate arrangements were purposefully designed to minimize optical crosstalk by the various fluorophores and test compounds. All measurements were performed at 25 °C in 60 μL of 50 mM tris, pH 8.0, dispensed into microplates via multichannel pipette. Compounds were shaken for 2 min on a plate shaker, centrifuged briefly for 1 min at $500 \times g$, then allowed to incubate at RT in light-reduced conditions for 10 min before measuring fluorescence intensity on a SpectraMax i3x plate reader under reduced lighting. Instrument settings: excitation filter wavelength (nm), emission filter wavelength (nm) with bandwidth filter widths in nm denoted in parentheses: AlexaFluor 350, 340 (15), 450 (15); fluorescein, 480 (15), 540 (25); AlexaFluor 488, 480 (15), 540 (25); resorufin, 525 (15), 598 (25); Texas Red, 547 (9), 618 (15); and AlexaFluor 647, 570 (9), 671 (15). Compound fluorescence intensity for fluorophores and test compounds was measured at each of the six fluorophore standard settings. Fluorophore standards present on each microplate were then used to construct normalized fluorescence concentration-responses ("fluorophore-equivalent concentrations", FEC) by nonlinear regression with 1/Y weighting. Lower limits of quantification (LLOQ) had >3:1 signal:noise ratio. Fluorescence intensities for each test compound were then converted to FECs. Calculated concentrations below the lower limits of quantification were scored as zero. Fluorophores prepared from independent dilutions as the calibrators were used as positive controls. Data are mean ± SD from three intra-run technical replicates performed on the same microplate.

## Quenching counter-screen
ngKATIs were assessed for fluorescence quenching using adaptions of previously published procedures[79]. Test compounds and individual

fluorophore standards prepared separately in assay buffer were incubated together, and the fluorescence intensity of these compound-fluorophore mixtures was compared to vehicle controls. Compounds were tested in triplicate at six final concentrations (32 nM to 100 μM via five-fold serial dilutions) at a fixed 250 nM fluorophore final concentration. All measurements were performed in 50 mM tris, pH 8.0 at 25 °C in 384-well black polystyrene microplates (Corning, cat # 3575) with 60 μL assay volumes. Compounds and fluorophore solutions were dispensed into microplates via multichannel pipette. Final concentration of DMSO was constant at 4.0% (v/v). Solutions were shaken for 2 min on a plate shaker, centrifuged briefly for 1 min at $500 \times g$, then incubated at RT in light-reduced conditions for 10 min before measuring fluorescence intensity on a SpectraMax i3x plate reader using the filter settings for each fluorophore. BHQ-10 carboxylic acid (LGC Biosearch Technologies, cat # BHQ-10-5) was used as a positive fluorescence quenching control compound. Significant fluorescence quenching was defined as signal reduction >25% of the corresponding fluorophore signal at any test compound concentration. Data are mean ± SD from three intra-run technical replicates performed on the same microplate.

### Immortalized cell-line histone acetylation assays

Select historical and next-generation KAT inhibitors were tested for their effects on cellular proliferation and H3K27ac levels in HEK293T and MCF7 cells as previously reported[31]. Cells were cultured in Dulbecco's Modified Eagle's Medium (DMEM) supplemented with 10% FBS (v/v; Winsent), penicillin (100 U mL$^{-1}$), and streptomycin (100 μg mL$^{-1}$). For cell growth analyses, cells were seeded in 96-well microplates, treated with the indicated compounds, and continuously monitored for 24 h using a live-cell Incucyte ZOOM imager (Essen Biosciences). Nuclei counts were determined using Vybrant DyeCycle Green (Invitrogen, cat # V35004, dilution 1:5000). Data are mean ± SD from three technical replicates performed on the same microplate.

For western blot analysis of H3K27ac, cells were treated for 24 h with compounds and lysed in ice-cold lysis buffer (20 mM tris-HCl, pH 8, 150 mM NaCl, 1 mM EDTA, 10 mM MgCl$_2$, 0.5% Triton X-100 (v/v), 12.5 U mL$^{-1}$ benzonase (Sigma, cat # E8263), complete EDTA-free protease inhibitor cocktail (Roche). After 3 min incubation, SDS was added to final 1% concentration (w/v). Total cell lysates were resolved using 4–12% bis-tris protein gels (Invitrogen) with MOPS buffer (Invitrogen) and transferred onto PVDF membranes (Millipore) in tris-glycine transfer buffer containing 10% MeOH (v/v) and 0.05% SDS (w/v). Membranes were blocked for 1 h in blocking buffer (5% milk in 0.1% Tween-20/PBS) and probed with the indicated primary antibodies overnight at 4 °C: H3K27ac (Cell Signaling Technologies, cat # 8173, dilution 1:1000), H3 (Abcam, cat # 10799, dilution 1:1000), and KAT3B (Bethyl, cat # A300-358A, dilution 1:2000). The following secondary antibodies were used according to manufacturer instructions: goat anti-rabbit IgG (IRDye 800-conjugated, LI-COR, cat # 926-32211, dilution 1:5,000) and donkey anti-mouse IgG (IRDye 680-conjugated, LI-COR, cat # 926-68072, dilution 1:5,000). The signal was acquired on an Odyssey scanner (LI-COR) at 800 and 700 nm. Antibody validation was provided by vendors (see "Reporting Summary").

Western blots in U-2 OS cells were performed similarly as above. Cells were cultured in DMEM (Thermo Fisher, cat # 10564011) supplemented with 10% FBS (v/v; Sigma, cat # F6178), penicillin (100 U mL$^{-1}$), and streptomycin (100 μg mL$^{-1}$), and maintained in a 37 °C, 5% CO$_2$ humidified incubator. Prior to compound treatment, 240,000 U-2 OS cells were plated per well in the above media in 6-well plates. The subsequent day, select compounds dissolved in neat DMSO were added to each well to the listed final concentration. The final DMSO concentration was maintained constant at 0.2% (v/v). After 24 h treatment, cells were harvested in 4x SDS-PAGE lysis buffer and boiled to denature. Cell lysates were separated on 16% tris-glycine gels (Thermo Fisher, cat # XP00165) for smaller proteins. For

larger proteins, lysates were separated on 3–8% tris-acetate gels (Thermo Fisher, cat # EA03785). All proteins were transferred to membranes via iBlot (Thermo Fisher). For KAT3B, membranes were blocked in 5% milk-TBS for 1 h, and then probed for KAT3B (Bethyl, cat # A300-358A, dilution 1:10,000) overnight in 2.5% milk-TBST (2.5% milk in 0.05% Tween-20/TBS). GAPDH (CST, cat # 2118S, dilution 1:2000) was also probed in 2.5% milk-TBST overnight at 4 °C. All other proteins were probed in 2.5% BSA-TBST (2.5% BSA in 0.1% Tween-20/TBS) overnight at 4 °C with the following dilutions: total H3 (Abcam, cat # AB1791, dilution 1:5000); H3K14ac (Millipore, cat # 07-353, dilution 1:2000); and H3K27ac (CST, cat # 8173, dilution 1:2000). All blots were probed with an anti-rabbit HRP conjugated secondary antibody (CST, cat # 7074S, dilution 1:5000) and washed with TBST (0.1% Tween-20/TBS) before visualization with enhanced chemiluminescence substrate. Antibody validation was provided by vendors (see Reporting Summary).

### Data analyses and figure preparation

All graphical data are expressed as mean ± standard deviation (SD) unless stated otherwise. Graphing and statistical analyses were performed using Prism (GraphPad, version 8.4.2) or *R* (version 3.6.1). Final figures were prepared in Adobe Illustrator (version 25.0).

### Statistics and reproducibility

No statistical method was used to predetermine sample size. No data were excluded from the analyses. The experiments were not randomized. The investigators were not blinded to allocation during experiments and outcome assessment.

For cell painting, the sample size of four technical replicates were chosen based on previous recommendations[7]. Compounds were also tested in concentration-response format (usually six concentrations). For other experiments, the number of technical replicates (usually three) are sufficient to determine significant differences between compounds within a high-throughput experiment. Select compounds were tested in multiple independent cell-painting experiments with acceptable reproducibility (Supplementary Fig. 3).

### Reporting summary

Further information on research design is available in the Nature Portfolio Reporting Summary linked to this article.

## Data availability

The multi-terabyte collection of CP images, metadata, and associated CellProfiler object-level files generated in this study have been deposited in the Image Data Resource database under accession code idr0133. The processed CP extracted feature data, the processed live-cell imaging data, the processed intracellular glutathione data, the raw ALARM NMR spectra, and the raw UPLC-MS data for KAT inhibitors have been deposited in the Figshare database under accession code 20293992. Source data are provided as a Source Data file. Key descriptors (categories, SMILES, purity, annotations) for study compounds and the composition of the proposed cellular injury informer set are provided in Supplementary Data 1. Source data are provided with this paper.

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

## Acknowledgements

Authors acknowledge Hayarpi Torosyan for technical assistance with AmpC purification; Brian Shoichet (sponsored by NIH R35GM122481) for help with aggregation controls; Maria Alimova and Jean Santos for assistance with CP image acquisition; Ryan Babcock, Allison Hands, Charlotte Sandland, and Anita Vrcic for compound management; Broad Pattern team for assistance with data visualization; and Liz Fuller for laboratory management. Authors acknowledge the following financial support: NIH NHLBI T32-HL007627 (JLD); National Science Foundation DGE1144152 and DGE1745303 (BKH); NIH NIGMS K99-GM124357 (BEZ); Harvard University's Graduate Prize Fellowship and Eli Lilly Graduate Fellowship in Chemistry (SDN); NIH NIGMS R37-GM62437 (PA Cole); NIGMS R35-GM127045 (SLS); Ono Pharma Breakthrough Science Initiative Award (BKW); NIH NIDDK U01-DK123717 (BKW); NIH NIGMS R35-GM122547 (SS). The authors gratefully acknowledge the use of the Opera Phenix imaging system at the Broad Institute, funded by the NIH S10 grant OD026839 (BKW). This research was supported in part by the Intramural/Extramural research program of the NCATS, NIH. The Structural Genomics Consortium is a registered charity (#1097737) that receives funds from AbbVie, Bayer Pharma AG, Boehringer Ingelheim, Canada Foundation for Innovation, Eshelman Institute for Innovation, Genome Canada through Ontario Genomics Institute [OGI-055], Innovative Medicines Initiative (EU/EFPIA) [ULTRA-DD grant no. 115766], Janssen, Merck & Co., Novartis Pharma AG, Ontario Ministry of Research, Innovation and Science (MRIS), Pfizer, São Paulo Research Foundation-FAPESP, Takeda, and the Wellcome Trust. The funders had no role in study design, data collection and analysis, decision to publish, or preparation of the manuscript. The opinions or assertions contained herein belong to the authors and are not necessarily the official views of the funders.

## Author contributions

Designed experiments: J.L.D., B.K.H. Performed KAT inhibitor interference profiling: J.L.D., J.H.S., M.C., F.L., M.C. Performed CP: J.L.D., S.D.N. Analyzed CP data: B.K.H., J.L.D., M.J.W., A.A., S.S., A.E.C., P.A. Clemons, B.K.W. Uploaded data to IDR: L.P.W.C., A.A. Performed DLS: P.L. Performed cellular histone profiling: B.E.Z., E.L.F., D.B.L., M.S.

Performed U-2 OS live-cell imaging and cellular health assays: J.L.D. Provided key reagents, supplies, or instrumentation: P.J.B., T.T., P.A. Cole, M.A.W., S.L.S., B.K.W. Authored the paper: J.L.D. Analyzed the data: J.L.D., B.K.H., B.K.W. Contributed with revisions: all authors.

## Competing interests

P.A. Cole is a cofounder of Acylin Therapeutics and has been a consultant to AbbVie, which have had research programs in KAT inhibitors. A.E.C. has ownership interest in, and serves on the Scientific Advisory Board of, Recursion, which uses image-based data for drug discovery. P.A. Clemons is an advisor for Pfizer, Inc., and Belharra Therapeutics. The other authors hereby declare no competing interests pertaining to the material in this manuscript.
