## [Peer Review File · Nature Communications]

Reviewers' Comments:

Reviewer #1:

Remarks to the Author:

In this manuscript, a collection of prototypical cytotoxic and nuisance compounds, along with a number of historical and next generation lysine acetyltransferase inhibitors (hKATi, ngKATi) were profiled with the Cell Painting assay, a target agnostic, imaging-based bioactivity screening approach. The biological activity observed in the Cell Painting assay was then compared to a variety of cell health and reactivity assays of various types. The authors observed that phenotypic profiles from the CP assay can distinguish compounds with distinct mechanisms of action and that a particular phenotypic cluster in the CP assay is strongly correlated with biological activity in other cell health assays. The end result of this work is a framework for establishing a nuisance compound informer set for HCS assays that would be useful in a variety of proposed applications (e.g. hit picking, HTS triage, mechanistic interpretation of unknown chemicals). The manuscript is well written, technically sound and presents results that will be of interest to the research community, in particular the pharmaceutical and toxicological research communities. Below are some minor suggestions that would improve the overall quality of the work.

- At several points in the manuscript, the authors use the phrase "chemical matter". To this reviewer's knowledge, this phrase is not commonly used in the scientific literature. What is meant by "chemical matter"? Is this different from just "chemicals" in some meaningful way?
- The authors begin the study by screening 218 "cytotoxins and prototypical nuisance compounds" but do not provide details on how these chemicals were initially selected. Please provide additional details on the selection criteria / process for these compounds, either in the manuscript text or as supplement.
- In Figure 6 the authors propose the idea of a nuisance compound informer set for use in interpretation of HCS assay results. Are the authors proposing that the 218 prototypical cytotoxicants and nuisance compounds they initially screened and used to construct phenotypes 1-9 is adequate for this purpose, a good starting point, etc. Or, should practitioners attempt to build their own informer sets based on the biological question at hand or cell model of interest? Some briefly clarifying language in the discussion would be helpful here.
- When visualized on PCA and compared to other clusters, the "gross injury" cluster (Cluster 9) appears to be somewhat of a catchall for cytotoxic treatments that produce profound morphological effects that don't look like anything in Clusters 1-8. That said, Cluster 9 still appears to be a useful classifier for readily identifying cytotoxic treatments. The "rainbow charts" such as those found in Figure 1f are an interesting and visually appealing way to readily understand the phenotypes associated with different concentrations of a test chemical. However, they are also a bit deceptive. Showing a series of test concentrations (points) connected by a line overlaid on a gradient ranging from 1 to 9 visually implies that the trajectory of the cellular phenotype has to progress through each stage before reaching the Cluster 9 "gross injury" phenotype. I don't think that is the case. The cells would not necessarily pass through each of the different phenotypic clusters as a function of dose. The authors should provide some clarifying language to the manuscript text or figure legend to ensure these plots are interpreted properly.
- Line 355: The ATCC acronym is misspelled.

Reviewer #2:

Remarks to the Author:

Summary

In this manuscript, Dahlin et al. presents their results towards identification of compounds causing cellular injury by the use of morphological profiling (Cell Painting). The authors analyse a historical cell painting dataset and finds that cell painting activity is correlated with loss of cells. Profiling a subset of 218 compounds in dose-response and clustering analysis reveals a cluster of compounds causing gross injury. Two compound categories are then the focus: Electrophiles and lysine

acetyltransferase inhibitors (KATIs). While non-specific electrophiles and historical KATIs cause broad cellular perturbations, resulting in cell painting profiles correlating to the gross injury cluster, targeted electrophiles, at low concentration, and next-generation KATIs show less cell painting activity and do not correlate to gross injury. The hKATIs and ngKATIs are extremely rigorously characterised in cell-free assay, revealing that ngKATIs are superior probe compounds for KATs. Next, 254 compounds were evaluated for modulation of "cell health" by profiling apoptosis induction (caspase activation), loss of confluence and membrane disruption (CytoTox) and this was correlated to activity in cell painting. High correlations between the cell health and cell painting activities were found, especially to compounds in "gross injury" cluster, indicating that cell painting can be used to determine interference with cell health. From this analysis the authors propose the generation of a compound collection, an informer set, that causes cellular injury, that can be used as control compounds in various phenotypic screens. By inference/correlation to the "cell injury compounds", compounds with unspecific activities can be down-prioritised as hits from HTS or HCS.

Assessment:

Overall, the work performed is thorough and provides a valuable resource for chemical biology researchers using the cell painting assay and for research and industry groups using HTS or HCS. The work shows that cell painting is good at detecting the perturbations related to cell injuries while more subtle effects, such as epigenetic changes, are not well detected, providing a clear perspective on what can be expected from cell painting. We recommend publication of this manuscript, but the authors can consider some improvements:

-The paper was hard to read and took significant effort to properly understand. There might be several different causes for this, but one is that the figures are extremely information-dense and, at least in the versions that we have received, has many features and fonts that simply cannot be distinguished. Furthermore, the text is very concise without much support for non-experts. The authors should very strongly consider the accessibility of the manuscript (Figures and text) to the more general reader. It is important that meaningful information can be readily extracted from all panels in the main figures. For instance, we had a hard time understanding the purpose of sub-panel 1c, but this also applies to other sub-panels. The figure-legends are likewise very brief on information. While it is a choice to leave the analysis of the presented data completely to the reader and just state the conclusion(s), providing a bit more details on how these conclusions are reached from the data would help guide the reader through the information presented. Again, this would make the paper more readable and approachable by non-experts, ultimately increasing its impact.

-Another issue that we believe the authors could provide a more nuanced discussion of relates to the general use of cell painting as a tool for generation of mechanistic hypotheses. The presented data confirms what has been speculated / an accepted fact in the community, namely that many bioactive compounds do not afford active cell painting profiles at the concentrations at which they should modulate their target. The authors specifically mention this on P8 line 197-199. This is actually a major take-home messages from the paper. Some additional comments or considerations from the authors along that direction could be favorably included in the discussion section. I.e. how large is, in their best view, the mechanistic space that CP, in its present form, can resolve in a meaningful way. Of course, as increasingly sophisticated analyses of image data, such as the fluorescence images that underlie CP, are being developed this may further improve the information that can be extracted and thereby the different phenotypes that can also be distinguished by CP.

Specific questions and comments:

-The following contains some more specific questions and comments, that should be clarified and/or changed before publication:

- Figure 1a, figure legend: "Active (Mahalanobis distance) CP compounds...", what is the threshold for mahalanobis distance? 3 SD's from mean? Is this a typical way of determining if a compound is active? It seems this would change with the dataset as the distribution of Mahalanobis distances would depend on the number of active and inactive compounds.

- Page 5 line 107 and Figure 1b: How are the clusters determined / distinguished? Hierarchical clustering does not directly give a number of optimal clusters. Is this by manual inspection of the correlation matrix in supplementary figure 1 or some algorithm? This should be included in the main text or in the methods section.
 - o Perhaps the clusters could be indicated in the correlation matrix in Fig S1
- Figure 1d: A non-specified cut-off is given for activity (Mahalanobis distance), perhaps 3 SD's as in figure 1a? As for figure 1a: Is it meaningful to have a cut-off for the activity measurement that depends on the subset of data analysed?
 - o The same cut-off is used in figure 2a and 3a it seems. Could the authors say if this is a general guideline?
- Page 5 line 120-122: The correlations from the existing MLI dataset to clusters 1-9 does not seem to be shown in figure 1e, as this contains retested compounds.
- Page 5 line 125: "We found that... were called bioactive upon retesting ...": What is the definition / threshold of "bioactive"? Is the mp-value by Hutz et al used or a Mahalanobis distance cut-off?
- Figure 1f: In rainbow plots: The dots have different sizes, does this mean anything? Seems like it could be correlated to CP activity score?
- Figure 2a: Is the legend for the activity score vs cell number and PCA plot the same? In the left panel, no description of point size is given, but colours are explained. Perhaps make shared legend between the two.
- Page 7, line 155: "...including lysine acetyltransferases (KATs), has recently ...", there seems to be missing a word like "inhibitors" or "binders"
- The compounds 468-472 are presented as ngKATI's, but it is not clear what 469 and 472 targets. Are they negative control compounds for 468 and 470+471 respectively? They seem inactive in the cell-free assays.
 - o It was noticed that 468 is compound A-485 and 469 is negative control compound called A-486 (from SGC). Perhaps this should be more clearly indicated in the manuscript
- Page 7 line 172: Reference to Fig S3 should probably be Fig S4 instead.
- Figure 3c: ngKATI's are called inactive (p7 line 177-178), but 468 and 471 are assigned to cluster 5, "kinase inhibitors" at some concentrations, even though they are inactive by the Mahalanobis distance. Does this reflect a real biological effect or is it by random chance? Are there high correlations to kinase inhibitors that could indicate a weak (off-target) activity?
- Figure 4a,b,c,: It is not clear what "overlap" in the rightmost plots is. There is no mention in the figure text or methods section as far as we can see.
- Figure 4d: A note: When printed on our printer, the light grey colour of the presumed "NA" cells (i.e. most 40 or 80 μ M datapoints) is identical to the white colour of the mean value. Perhaps a different colour can be chosen for "NA". This meaning of the grey values should also be indicated in the figure text.
- A note: Besides use for sorting out nuisance compounds through their broad effects on cell health, which is the main focus of the current study, a related perspective of CP is its use to study the differential activity of compounds that display both antibacterial activity as well as unspecific effects on mammalian cells. The idea being to prioritise those compounds for further studies/development as potential antibiotics that display the smallest degree of perturbation of mammalian cell health. Nat. Chem. 2021, 13, 47-55.
- Page 10 and line 236-237: "We found a strong correlation between compound treatments with strong CP signals..." How is a strong CP signal defined? Mahalanobis distance threshold or other measure?

- Figure 5b: It is very hard (impossible) to distinguish the activity scores by the tone of grey when the points are overlapping and are semi-transparent. Perhaps different colours or point shapes could be used, or the stratification could be removed.
- Page 10 line 237-241 and Figure 5: The reference to panels in the text seems to be off, e.g. the text refers to the entanglement in figure 5c, but it is in panel d. There is no reference to panel e or f in the text.
- The authors propose the use of a 'cell injury informer set'. It seems the list is in the excel supporting file (informer set column, denoted by "Y"), but this does not seem to be referenced in the manuscript. Page 10 line 255: "... we propose an informer set of control com-pounds to model cell injury ...", the list should be referenced here?
- Page 18, line 424-436: Is the Mahalanobis distance calculated before feature reduction?
- Supplementary note 1: Rule-Of-Five compliance references Fig S3. Perhaps this should be Fig S4?

Reviewer #3:

Remarks to the Author:

The authors have generated a novel resource for possible detection of non-specific bioactivity in high-throughput phenotypic and high-content assays. The Resource includes 218 "prototypical cytotoxic and nuisance compounds" that have been tested in the "Cell Painting" (CP) phenotypic assays. The authors suggest that these compounds "provide a blueprint for routinely detecting nuisance compounds in triage activities during HTS". Overall, this is an interesting study, and its results may be valuable for the entire field of phenotypic assays.

The experiments are described in great detail and the data generated in this study are shared with the community including a large collection (11 TB) of images in a web-accessible database, which is certainly a plus of this study. What is somewhat questionable is the breadth of the appeal and whether the reported observations and claims of the study are in sync. Specifically, the authors seem to suggest two major applications of their data: (i) the library of 218 "trouble-maker" compounds that can be used to test an assay robustness, and (ii) the reference profiles of these compounds in the CP assay that can be used to detect if a new molecule is a nuisance compound. The big question is whether the reported data can be extrapolated to other assays or other compounds, and whether the signal to noise ratio reported in this study is high enough for the task.

If the authors agree with the above summary of their chief messages, then in this reviewer's opinion, these messages are not outlined crisply or quantitatively; so, it is recommended to do so in the revised manuscript. Specifically, in can the authors suggest specific assays (other than CP) where they expect this benchmark set of 218 compounds (also, how this specific set was selected?) to perform well? And if a new set of compounds is profiled in CP (or other assays) can the authors forecast the expected accuracy of determining if a compound can be classified as a nuisance compound based on its profile?

There are several additional comments or questions as follows.

- Until recently, it was popular to predict nuisance compounds using structural alters (e.g., PAINS). Have the authors attempted to use such predictors for the compound library they selected for testing?

- Line 55: "The utility of certain compound classes, including lysine acetyltransferases (KATs), has recently been questioned...": KATs are not compound classes; the authors probably meant KATIs.

- Line 181: "The ngKATIs occupied different PCA feature-space from most hKATIs, with the summary morphological fingerprints being essentially null for ngKATIs while the hKATIs mirrored cluster 9 (Figure 3b)." Figure 3B shows that hKATIs are more distributed in the PC space but a good fraction of them forms a cluster nearly overlapping with ngKATIs' cluster; so, the distinctiveness of these two classes based on this analysis is not very obvious. Can the authors comment on this observation.

- Line 202: "Other ngKATIs likely behave similarly, given some shared chemical scaffolds and the

lack of red-flag interference chemotypes^{38,39}." Can the authors provide more chemical structure sensitive information, i.e., what shared scaffolds and how prevalent are "red-flag" phenotypes are in hKATIs vs ngKATIs as well as in other compounds in their dataset? Is there a correlation between chemotypes and nuisance behavior?

- Line 208: "The CP activities and relative cell numbers of 254 profiled compounds...": Where does this number of compounds come from? Prior to this, the authors were describing a 218 compound dataset.

Reviewer #4:

Remarks to the Author:

In Dahlin et al., a generalizable cellular imaging approach and resource are provided for flagging nuisance compounds and prioritizing safer chemotypes from phenotypic discovery. The work is accompanied by publicly available cellular profiling images in U2OS. The authors studied 218 compounds in dose response by Cell Painting, a well-established technique from the same authors, and also compared the results to a related MLI dataset. Good correlation of the 2 datasets was found for many of the MLI compounds, but not all. I don't recall seeing any speculation around the compounds that were not correlated, this may be useful to add for some compounds even if anecdotally for one or two.

The authors note strong connection between cell death/depletion and the CP bioactivity score, as well as other key markers of cellular imaging in live profiling.

Important observations were detailed around the improved next gen KATIs vs historical compounds. Similarly, non-specific electrophiles fared worse than targeted covalent drugs, including glutathione alterations, although notably in some cases the compound target wasn't present in U2OS (KRAS G12C, BTK). This is important because, for example, ibrutinib is noted to have drug-induced liver injury in some patients, and such a generalized approach would miss more tissue specific tox effects of drugs. It may be especially of interest for drug R&D to develop CP protocols in hepatocytes, cardiomyocytes, and other cell types more connected to safety profiling downstream of lead characterization.

24 hr and 48 hr timepoint comparisons showed 24 hr time points may often be used for efficiency's sake.

A couple of suggestions:

In Fig 1c, I can't read the cellular features text due to the colored columns and small font when reading the PDF version with figures positioned vertical. If the journal is willing to present Fig 1 horizontally on the page, this could be ameliorated.

Figures 5 c/d/e/f- please check that the corresponding text in the manuscript corresponds with these panels appropriately and that all panels are referenced.

Finally- the key output of this resource paper for me is the list of suggested compounds covering various tox mechanisms. While Fig 6 gives an exemplar approach, I assume this is provided in Supplemental XLS File 1. However, it wasn't clear to me if the authors intended Column F "Informer Set" as their recommended set for anyone intending to follow this up. There is also Column D "Cellular injury" which is nice too. But I would really like it stated somewhere what the recommended set is, given that's the crux of the paper.

06 October 2022

Dear Dr. Eldridge,

We thank you for the review decision for our manuscript NCOMMS-22-30998-T. We have enclosed a revised version of our manuscript entitled, "Reference compounds for characterizing cellular injury in high-content cellular morphology assays" for consideration for publication in Nature Communications.

To assist with reviewing, we have also included a "tracked changes" version of our manuscript.

The following are our responses to reviewer feedback:

REVIEWER #1

In this manuscript, a collection of prototypical cytotoxic and nuisance compounds, along with a number of historical and next generation lysine acetyltransferase inhibitors (hKATi, ngKATi) were profiled with the Cell Painting assay, a target agnostic, imaging-based bioactivity screening approach. The biological activity observed in the Cell Painting assay was then compared to a variety of cell health and reactivity assays of various types. The authors observed that phenotypic profiles from the CP assay can distinguish compounds with distinct mechanisms of action and that a particular phenotypic cluster in the CP assay is strongly correlated with biological activity in other cell health assays. The end result of this work is a framework for establishing a nuisance compound informer set for HCS assays that would be useful in a variety of proposed applications (e.g. hit picking, HTS triage, mechanistic interpretation of unknown chemicals). The manuscript is well written, technically sound and presents results that will be of interest to the research community, in particular the pharmaceutical and toxicological research communities. Below are some minor suggestions that would improve the overall quality of the work.

Author response: We thank the Reviewer for this overall feedback. We have addressed these minor suggestions as noted below.

At several points in the manuscript, the authors use the phrase "chemical matter". To this reviewer's knowledge, this phrase is not commonly used in the scientific literature. What is meant by "chemical matter"? Is this different from just "chemicals" in some meaningful way?

Author response: This phrase is more common in the medicinal chemistry and drug discovery literature, but there is no reason not to use "chemicals" to avoid confusion. Accordingly, we have changed the phrase "chemical matter" to "chemicals" or "compounds" in our revised manuscript. We thank the Reviewer for this suggestion.

The authors begin the study by screening 218 "cytotoxins and prototypical nuisance compounds" but do not provide details on how these chemicals were initially selected. Please provide additional details on the selection criteria / process for these compounds, either in the manuscript text or as supplement.

Author response: We agree with the Reviewer that these are important details for the reader. Therefore, we have included additional details summarizing our compound selection process as a Supplementary Note in our revised manuscript ("Supplementary Note 1. Summary of compound selection process").

In Figure 6 the authors propose the idea of a nuisance compound informer set for use in interpretation of HCS assay results. Are the authors proposing that the 218 prototypical cytotoxicants and nuisance compounds they initially screened and used to construct phenotypes

1-9 is adequate for this purpose, a good starting point, etc. Or, should practitioners attempt to build their own informer sets based on the biological question at hand or cell model of interest? Some briefly clarifying language in the discussion would be helpful here.

Author response: We have included a more explicit reference to the formulation of the proposed cellular injury informer set in our revised manuscript. The text now states: “Based on our data and cumulative experience with HTS, we propose an informer set of control compounds to model cell injury phenotypes in HCS and other phenotypic assays including mechanism-based and nonspecific modes of gross cellular injury (Figure 6; Supplementary Data 1, column “Proposed informer set”).”

We agree with the Reviewer that additional clarifying language would be useful for readers. To further clarify these points, we have included the following text in the discussion section of our revised manuscript: “This proposed set should serve as a useful starting point for practitioners, and could be subject to future improvements as additional evidence is generated by the scientific community. Although practitioners could build their own custom sets, using a common set (or even subset) of reference cellular injury compounds may benefit the scientific community as a whole and enable data harmonization. It is likely that modifications to the set may be needed for specific assays and model systems, such as the addition of compounds with other cytotoxic MoAs, or the removal of redundant compounds if throughput or cost is a concern.”

When visualized on PCA and compared to other clusters, the “gross injury” cluster (Cluster 9) appears to be somewhat of a catchall for cytotoxic treatments that produce profound morphological effects that don’t look like anything in Clusters 1-8. That said, Cluster 9 still appears to be a useful classifier for readily identifying cytotoxic treatments. The “rainbow charts” such as those found in Figure 1f are an interesting and visually appealing way to readily understand the phenotypes associated with different concentrations of a test chemical. However, they are also a bit deceptive. Showing a series of test concentrations (points) connected by a line overlaid on a gradient ranging from 1 to 9 visually implies that the trajectory of the cellular phenotype has to progress through each stage before reaching the Cluster 9 “gross injury” phenotype. I don’t think that is the case. The cells would not necessarily pass through each of the different phenotypic clusters as a function of dose. The authors should provide some clarifying language to the manuscript text or figure legend to ensure these plots are interpreted properly.

Author response: This is an excellent point, and we thank the Reviewer for this feedback. Our original intention for connecting the dots is to provide readers with a visual guide to track phenotypic progression, but we agree that this (along with the numbering of clusters) could imply to some readers that the trajectory of the cellular phenotype has to progress through each cluster before reaching the Cluster 9 “gross injury” phenotype.

We have included the following text in the Figure 1 legend of our revised manuscript: “Select CP profiles of cellular injury compounds; rainbow plots denote assigned cluster at each compound concentration; arrow indicates compound concentration of representative image. For rainbow plots, note that phenotypic trajectories do not have to progress through each cluster before reaching the cluster 9 “gross injury” phenotype (dotted lines).”

Furthermore, we have changed the lines connecting the points in these rainbow plots (Figures 1-3) to dotted lines. We hope that this will further indicate that the trajectory is assumed.

Line 355: The ATCC acronym is misspelled.

Author response: We have corrected the misspelled acronym in our revised manuscript.

REVIEWER #2

Summary

In this manuscript, Dahlin et al. presents their results towards identification of compounds causing cellular injury by the use of morphological profiling (Cell Painting). The authors analyse a historical cell painting dataset and finds that cell painting activity is correlated with loss of cells. Profiling a subset of 218 compounds in dose-response and clustering analysis reveals a cluster of compounds causing gross injury. Two compound categories are then the focus: Electrophiles and lysine acetyltransferase inhibitors (KATIs). While non-specific electrophiles and historical KATIs cause broad cellular perturbations, resulting in cell painting profiles correlating to the gross injury cluster, targeted electrophiles, at low concentration, and next-generation KATIs show less cell painting activity and do not correlate to gross injury. The hKATIs and ngKATIs are extremely rigorously characterised in cell-free assay, revealing that ngKATIs are superior probe compounds for KATs. Next, 254 compounds were evaluated for modulation of "cell health" by profiling apoptosis induction (caspase activation), loss of confluence and membrane disruption (CytoTox) and this was correlated to activity in cell painting. High correlations between the cell health and cell painting activities were found, especially to compounds in "gross injury" cluster, indicating that cell painting can be used to determine interference with cell health. From this analysis the authors propose the generation of a compound collection, an informer set, that causes cellular injury, that can be used as control compounds in various phenotypic screens. By inference/correlation to the "cell injury compounds", compounds with unspecific activities can be down-prioritised as hits from HTS or HCS.

Author response: We thank the Reviewer for this overall feedback and believe this is an accurate summary of the manuscript.

Assessment:

Overall, the work performed is thorough and provides a valuable resource for chemical biology researchers using the cell painting assay and for research and industry groups using HTS or HCS. The work shows that cell painting is good at detecting the perturbations related to cell injuries while more subtle effects, such as epigenetic changes, are not well detected, providing a clear perspective on what can be expected from cell painting. We recommend publication of this manuscript, but the authors can consider some improvements:

Author response: We thank the Reviewer for this overall feedback. We have addressed these minor suggestions as noted below.

The paper was hard to read and took significant effort to properly understand. There might be several different causes for this, but one is that the figures are extremely information-dense and, at least in the versions that we have received, has many features and fonts that simply cannot be distinguished. Furthermore, the text is very concise without much support for non-experts. The authors should very strongly consider the accessibility of the manuscript (Figures and text) to the more general reader. It is important that meaningful information can be readily extracted from all panels in the main figures. For instance, we had a hard time understanding the purpose of sub-panel 1c, but this also applies to other sub-panels. The figure-legends are likewise very brief on information. While it is a choice to leave the analysis of the presented data completely to the

reader and just state the conclusion(s), providing a bit more details on how these conclusions are reached from the data would help guide the reader through the information presented. Again, this would make the paper more readable and approachable by non-experts, ultimately increasing its impact.

Author response: We thank the Reviewer for this valuable feedback. We have addressed the readability issues by several means: (1) we have included high-resolution versions of our figures in our revised submission in case the poor readability was due to a technical issue, (2) we have significantly expanded the figure captions, (3) we have included additional text throughout the revised the results section to provide readers about how certain conclusions are reached based on the data. We believe these changes, along with the other revisions such as expanded commentary in the discussion section, should improve the readability of our revised manuscript without adding excessive length.

Another issue that we believe the authors could provide a more nuanced discussion of relates to the general use of cell painting as a tool for generation of mechanistic hypotheses. The presented data confirms what has been speculated / an accepted fact in the community, namely that many bioactive compounds do not afford active cell painting profiles at the concentrations at which they should modulate their target. The authors specifically mention this on P8 line 197-199. This is actually a major take-home messages from the paper. Some additional comments or considerations from the authors along that direction could be favorably included in the discussion section. I.e. how large is, in their best view, the mechanistic space that CP, in its present form, can resolve in a meaningful way. Of course, as increasingly sophisticated analyses of image data, such as the fluorescence images that underlie CP, are being developed this may further improve the information that can be extracted and thereby the different phenotypes that can also be distinguished by CP.

Author response: We thank the Reviewer for raising this issue. We have included the following additional text in the discussion section of our revised manuscript: “An important observation from testing the historical and next-generation KAT inhibitors are that some high-quality bioactive compounds do not lead to active CP profiles at concentrations at which they modulate their target. This suggests there are limits to the mechanistic space that can be captured by CP. In some cases, bioactive compounds may simply not produce detectable morphological changes in cells, whether at all or within the usual 24- to 48-h compound treatment windows of the conventional CP assay. Therefore, it should not be assumed that a bioactive compound will produce a CP phenotype. More sophisticated image analyses and/or alternative experimental protocols may improve the number and type of phenotypes that can be reliably detected with CP and similar morphological assays. The large number of phenotypes caused by cellular injury compounds (cluster 9) suggests a significant portion of CP morphologies may be affected by compound-mediated cellular injury. For any given morphological assay, this could be estimated by testing a diverse selection of bioactive probes and cellular injury control compounds.”

Figure 1a, figure legend: “Active (Mahalanobis distance) CP compounds...”, what is the threshold for mahalanobis distance? 3 SD's from mean? Is this a typical way of determining if a compound is active? It seems this would change with the dataset as the distribution of Mahalanobis distances would depend on the number of active and inactive compounds.

Author response: The threshold for “active” is 3 SD from the mean, as denoted in Figure 1A by the dotted line and the coloring of the “active” box. The Mahalanobis distances are calculated with respect to the vehicle-treated well profiles, i.e. DMSO-treated wells. The goal of this metric is to provide a measure of how different a compound’s effect is on cell morphology compared to the effect of DMSO. It is prudent to make this comparison for each assay plate as well as for each cell-painting experiment to account for any plate effects or day-to-day variability. In our analysis of this historical data, the definition of “active” depends on the behavior of the DMSO-treated wells, not the number of active/inactive compounds.

Hutz *et al.* suggest that both mp-value and Mahalanobis distance are reasonable metrics to consider, and that in fact mp-value is not usually optimal for compound prioritization. This prioritization effort benefits from inspection of not only whether a treatment is distinct from vehicle control, but also how large the difference is. In that same vein we chose a cutoff of the activity score to flag compounds as "active."

Relevant text from Hutz et al: In some cases, such as in the HCS data set tested here, nearly all of the assayed treatments may have statistically significant mp-values. The mp-value itself is designed to merely say whether two treatments are different; its value should not be used to further prioritize this subset of significant treatments. However, as mentioned above, the Mahalanobis distance calculated during the mp-value calculation process can be used for a secondary prioritization, as it indicates the magnitude of the difference between the treatments.

Page 5 line 107 and Figure 1b: How are the clusters determined / distinguished? Hierarchical clustering does not directly give a number of optimal clusters. Is this by manual inspection of the correlation matrix in supplementary figure 1 or some algorithm? This should be included in the main text or in the methods section. Perhaps the clusters could be indicated in the correlation matrix in Fig S1

Author response: The choice of nine clusters was based on several competing factors. Too few clusters would not be specific enough for certain MoAs, while an excessive number of clusters could may lead to overfitting. Given our past experiences with cell painting, we also examined the clustering of microtubule poisons, which produce a characteristic and highly reproducible phenotype/cluster. Additionally, since all of the images and associated data are publicly available, readers can experiment with alternative numbers of clusters.

The clusters in the correlation matrix in Figure S1 are not the same as for the rest of the paper. To avoid confusion, we have now updated the corresponding figure caption to state: “Compounds (1-171, 455-501) were sorted by unsupervised hierarchical clustering using the average profile for each compound across all treatment concentrations “ For the rest of the paper, the doses are considered (and clustered) separately. The main reason was to provide a more visually tractable heatmap for inspection (there would be six times as many rows and columns if not for the aggregation).

Figure 1d: A non-specified cut-off is given for activity (Mahalanobis distance), perhaps 3 SD’s as in figure 1a? As for figure 1a: Is it meaningful to have a cut-off for the activity measurement that depends on the subset of data analysed? The same cut-off is used in figure 2a and 3a it seems. Could the authors say if this is a general guideline?

Author response: As in Figure 1D, the threshold for “active” is 3 SD from the mean using the the Mahalanobis distances calculated with respect to the vehicle-treated well profiles, i.e. DMSO-treated wells. The goal of this metric is to provide a measure of how different a compound’s effect is on cell morphology compared to the effect of DMSO.

To make this more explicit, we have revised the figure caption. Our revised manuscript now states: “Clusters of cell injury compounds correlate with cell number. The cut-off is 3 SD from the mean of DMSO-treated wells using the CP activity (Mahalanobis distances).” We have also revised the figure panels (Figure 1D, 2A, 3A) to note the cut-off is 3 SD.

Page 5 line 120-122: The correlations from the existing MLI dataset to clusters 1-9 does not seem to be shown in figure 1e, as this contains retested compounds.

Author response: The correlations of the existing MLI dataset are in Figure 1E. This is noted in the figure caption: “Inset: heatmap and dendrogram shows pairwise correlation coefficients between each MLI CP compound profile and each of the 9 clusters (red arrowhead, enrichment of cluster 9)”.

Page 5 line 125: “We found that... were called bioactive upon retesting ...”: What is the definition / threshold of “bioactive”? Is the mp-value by Hutz et al used or a Mahalanobis distance cut-off?

Author response: The Mahalanobis distance cut-off was used; the mp-value by Hutz. et al. was not used. We have indidcarted this in our revised manuscript as follows: “We found that, upon retesting, 98/119 (82%) and 21/166 (13%) of MLI-HC and MLI-NC compounds were called bioactive upon retesting based on Mahalanobis distances.”

Figure 1f: In rainbow plots: The dots have different sizes, does this mean anything? Seems like it could be correlated to CP activity score?

Author response: Yes, the dot sizes in the rainbow plots are sized according to their CP activity score. We have included a legend to indicate this sizing in our revised Figure 1.

Figure 2a: Is the legend for the activity score vs cell number and PCA plot the same? In the left panel, no description of point size is given, but colours are explained. Perhaps make shared legend between the two.

Author response: We thank the Reviewer for this helpful suggestion. Our revised Figure 2A and Figure 3A now contain shared figure legends.

Page 7, line 155: “..including lysine acetyltransferases (KATs), has recently ...”, there seems to be missing a word like “inhibitors” or “binders”

Author response: We have corrected this text in our revised manuscript. It now states “The utility of certain compound classes, including lysine acetyltransferase (KAT) inhibitors, has recently been questioned.”

The compounds 468-472 are presented as ngKATI's, but it is not clear what 469 and 472 targets. Are they negative control compounds for 468 and 470+471 respectively? They seem inactive in

the cell-free assays. It was noticed that 468 is compound A-485 and 469 is negative control compound called A-486 (from SGC). Perhaps this should be more clearly indicated in the manuscript

Author response: We have more clearly indicated that 469 and 472 are negative control analogs in our revised manuscript. In the main text, it now states: “However, highly potent and specific “next-generation” KAT inhibitors (ngKATIs) have now been reported, including the KAT3 inhibitor A-485 (468) and its negative control analog A-486 (469), and the KAT6 inhibitors WM-8014 (470) and WM-1119 (471), and their negative control analog WM-2474 (472).”

Page 7 line 172: Reference to Fig S3 should probably be Fig S4 instead.

Author response: We have corrected this incorrect figure reference in our revised manuscript.

Figure 3c: ngKATI's are called inactive (p7 line 177-178), but 468 and 471 are assigned to cluster 5, “kinase inhibitors” at some concentrations, even though they are inactive by the Mahalanobis distance. Does this reflect a real biological effect or is it by random chance? Are there high correlations to kinase inhibitors that could indicate a weak (off-target) activity?

Author response: After we examined the correlation for each compound at each concentration to cluster 5 (kinase cluster), we speculate that this is due to random chance, though we cannot rule out some weak off-target activity.

This situation illustrates the importance of perform follow-up experiments. In this specific case, one could perform cellular kinome profiling to investigate whether this cluster assignment is in fact due to some off-target kinase activity.

Figure 4a,b,c,: It is not clear what “overlap” in the rightmost plots is. There is no mention in the figure text or methods section as far as we can see.

Author response: In Figure 4, “overlap” refers to the number of objects (cells) that are positive for caspase 3/7 activation (green fluorescence) and loss of membrane integrity (red fluorescence). Like confluence, caspase 3/7 activation, and cell viability (membrane integrity), these values were then calculated as an AUC to account for testing in concentration-response format. We have now included a description of the overlap readout in the figure legend and methods sections of our revised manuscript. In the accompanying supplemental data, we have also quantified the overlap in terms of area.

Figure 4d: A note: When printed on our printer, the light grey colour of the presumed “NA” cells (i.e. most 40 or 80 μ M datapoints) is identical to the white colour of the mean value. Perhaps a different colour can be chosen for “NA”. This meaning of the grey values should also be indicated in the figure text.

Author response: We thank the Reviewer for this useful feedback. We have re-colored the NA values as solid black, and indicated this in the figure and figure legend. We have recolored outlier values as solid yellow (i.e., outside range, non-calculable GSG:GSSG ratios), and indicated this in the figure legend. Note that these data are freely available in the referenced Figshare directory should readers wish to know the exact values.

A note: Besides use for sorting out nuisance compounds through their broad effects on cell health, which is the main focus of the current study, a related perspective of CP is its use to study the differential activity of compounds that display both antibacterial activity as well as unspecific effects on mammalian cells. The idea being to prioritise those compounds for further studies/development as potential antibiotics that display the smallest degree of perturbation of mammalian cell health. Nat. Chem. 2021, 13, 47-55.

Author response: We thank the Reviewer for raising this additional perspective regarding cell painting. This approach could likely benefit from purposefully including nuisance/cytotoxic compounds to characterize cellular injury phenotypes that would want to be avoided during antibiotic discovery. We have included a reference to this work in our revised manuscript, since it helps to further articulate the potential scope of our work.

Our revised text now states: “Groups have developed customized assays to detect nephrotoxicity, pulmonotoxicity, antibiotic toxicity in mammalian cells, and other toxicities using specialized cell models and stains for each.”

Page 10 and line 236-237: “We found a strong correlation between compound treatments with strong CP signals...” How is a strong CP signal defined? Mahalanobis distance threshold or other measure?

Author response: We have revised this text to indicate our cutoff for strong CP signal. Our revised manuscript now states: “There was strong correlation between replicate compound treatments at each of 24- and 48-h, and high correlation between the pair-wise 24- and 48-h compound treatments (Figure 5a). There was also a strong correlation between compound treatments with strong CP signals (i.e., CP

activity/Mahalanobis distance > 10), which could be attributed to the higher signal-to-noise of their CP profiles (Figure 5b).”

Furthermore, we have revised the coloring of Figure 5b to make the link between high CP activity and strong correlation qualitatively more apparent (see next response).

Figure 5b: It is very hard (impossible) to distinguish the activity scores by the tone of grey when the points are overlapping and are semi-transparent. Perhaps different colours or point shapes could be used, or the stratification could be removed.

Author response: We thank the Reviewer for pointing this out. Our revised Figure 5B now indicates the activity score by color gradient.

Page 10 line 237-241 and Figure 5: The reference to panels in the text seems to be off, e.g. the text refers to the entanglement in figure 5c, but it is in panel d. There is no reference to panel e or f in the text.

Author response: We thank the Reviewer for alerting us to these errors. We have corrected the figure references in our revised manuscript.

The authors propose the use of a ‘cell injury informer set’. It seems the list is in the excel supporting file (informer set column, denoted by “Y”), but this does not seem to be referenced in the manuscript. Page 10 line 255: “... we propose an informer set of control compounds to model cell injury ...”, the list should be referenced here?

Author response: We have included a reference to the formulation of the proposed cellular injury informer set in our revised manuscript. The text now states: “Based on our data and cumulative experience with HTS, we propose an informer set of control compounds to model cell injury phenotypes in HCS and other phenotypic assays including mechanism-based and nonspecific modes of gross cellular injury (Figure 6; Supplementary Data 1, column “Proposed informer set”).”

We have also modified the column header in the revised supporting file to better indicate the compounds we propose as part of a cellular injury informer set (“Proposed informer set”).

Furthermore, we have modified our description of the Supplementary materials to now state: “Supplementary Data 1: Key compound descriptors (categories, SMILES, purity, annotations) for study compounds and proposed cellular injury informer set (XLSX).”

Page 18, line 424-436: Is the Mahalanobis distance calculated before feature reduction?

Author response: Yes, the Mahalanobis distances are calculated before feature reduction, although it is preceded by dimensionality reduction via PCA and taking the first principal components capable of explaining $\geq 90\%$ of the variance (outlined in Methods section).

Apart from the Mahalanobis-distance calculations, additional feature reduction was required to avoid the over-representation of highly similar cell-painting features, in particular for downstream analyses that dealt with compound-compound correlations, like the hierarchical clustering. To clarify this feature reduction after calculating the

Mahalanobis distances, we have amended our Methods section to state: “The R *cytominer* package was then used to reduce the number of redundant features by removing those which were highly correlated.”

Supplementary note 1: Rule-Of-Five compliance references Fig S3. Perhaps this should be Fig S4?

Author response: Correct, and we have corrected this figure reference in our revised manuscript.

REVIEWER #3

The authors have generated a novel resource for possible detection of non-specific bioactivity in high-throughput phenotypic and high-content assays. The Resource includes 218 “prototypical cytotoxic and nuisance compounds” that have been tested in the “Cell Painting” (CP) phenotypic assays. The authors suggest that these compounds “provide a blueprint for routinely detecting nuisance compounds in triage activities during HTS”. Overall, this is an interesting study, and its results may be valuable for the entire field of phenotypic assays. The experiments are described in great detail and the data generated in this study are shared with the community including a large collection (11 TB) of images in a web-accessible database, which is certainly a plus of this study. What is somewhat questionable is the breadth of the appeal and whether the reported observations and claims of the study are in sync. Specifically, the authors seem to suggest two major applications of their data: (i) the library of 218 “trouble-maker” compounds that can be used to test an assay robustness, and (ii) the reference profiles of these compounds in the CP assay that can be used to detect if a new molecule is a nuisance compound. The big question is whether the reported data can be extrapolated to other assays or other compounds, and whether the signal to noise ratio reported in this study is high enough for the task. If the authors agree with the above summary of their chief messages, then in this reviewer’s opinion, these messages are not outlined crisply or quantitatively; so, it is recommended to do so in the revised manuscript. Specifically, in can the authors suggest specific assays (other than CP) where they expect this benchmark set of 218 compounds (also, how this specific set was selected?) to perform well? And if a new set of compounds is profiled in CP (or other assays) can the authors forecast the expected accuracy of determining if a compound can be classified as a nuisance compound based on its profile?

Author response: We thank the Reviewer for this valuable feedback. To clarify, we actually suggest a subset of the 218 profiled compounds to be used as a reference set for phenotypic assays. As addressed in other reviewer comments, we have made this more explicit in our revised manuscript. We have also specified our selection process for the compound set as a supplementary note.

Our data suggest that cellular injury compounds should produce sufficiently high signal:noise to make them robust choices for cell painting reference compounds. Furthermore, our supplementary data show that these phenotypes are reproducible in independent experiments. Based on our experiences with phenotypic screening and gene expression profiling (including L1000), we hypothesize that this set should be applicable to a wide variety of screening and profiling assays, and not exclusive to just cell painting. This is supported by the observations that cell injury compounds produce strong CP phenotypes and strong L1000 signatures, and the targets/pathways/mechanisms for cell injury are inherent to most biological systems.

The question about forecasting accuracy and whether a compound can be predicted as a nuisance is currently unsettled. As we have previously described, cellular nuisance behavior is highly context dependent and is more difficult to neatly classify than biochemical nuisance behavior (PMID 33592188).

We have included the following additional text in the discussion section of our revised manuscript to address the points regarding the applicability to other assays, and our predictions regarding expected accuracy:

- “Although we only profiled one cell line, this approach is likely generalizable to other biological systems and profiling assays”
- “In one study, compounds profiled with the L1000 transcriptome profiling assay and CP, cytotoxic compounds produced robust signatures in both techniques⁴⁶. This further suggests that the proposed approach can be applied to other assays and cell types.”
- “Given the complexities of cellular nuisance compounds and their dependence on context, it is difficult at this point to quantify the sensitivity and accuracy of such an informer set in predicting whether an active compound is acting by a nuisance mechanism. The use of such a standardized set by the chemical biology and drug discovery communities should help to address this important question.”

Until recently, it was popular to predict nuisance compounds using structural alerts (e.g., PAINS). Have the authors attempted to use such predictors for the compound library they selected for testing?

Author response: We did not explicitly use structural alerts (such as PAINS) for this compound library. Since our overall goal was to characterize the effect of cytotoxic compounds on the cell painting readout, we wanted to include a broad class of chemotypes and cytotoxic mechanisms. PAINS, for example, were derived from cell-free AlphaScreen assays, and in our collective experience are generally enriched in nonspecific electrophiles. In our proposed informer set, nonspecific electrophiles are accounted for by several of the historical KAT inhibitors.

Line 55: “The utility of certain compound classes, including lysine acetyltransferases (KATs), has recently been questioned...”: KATs are not compound classes; the authors probably meant KATIs.

Author response: We have corrected this text in our revised manuscript. It now states “The utility of certain compound classes, including lysine acetyltransferase (KAT) inhibitors, has recently been questioned.”

Line 181: “The ngKATIs occupied different PCA feature-space from most hKATIs, with the summary morphological fingerprints being essentially null for ngKATIs while the hKATIs mirrored cluster 9 (Figure 3b).” Figure 3B shows that hKATIs are more distributed in the PC space but a good fraction of them forms a cluster nearly overlapping with ngKATIs’ cluster; so, the distinctiveness of these two classes based on this analysis is not very obvious. Can the authors comment on this observation.

Author response: This observation is mostly a reflection of the concentration response. The hKATIs that overlap the ngKATI cluster in this PCA plot correspond to the hKATIs tested at lower compound concentrations (below the concentrations were

they appear to be cell-active). By contrast, the hKATIs points that do not overlap with the ngKATI cluster in this PCA plot (and occupy cluster 9) are at the relatively higher concentrations where they decrease histone acetylation.

To clarify this point, our revised manuscript now states: “The ngKATIs occupied different PCA feature-space from most hKATIs, with the latter occupying cluster 9 (cell injury) when tested at higher concentrations that coincide with their reported cellular KAT inhibition activities (Figure 3a).”

Line 202: “Other ngKATIs likely behave similarly, given some shared chemical scaffolds and the lack of red-flag interference chemotypes^{38,39}.” Can the authors provide more chemical structure sensitive information, i.e., what shared scaffolds and how prevalent are “red-flag” phenotypes are in hKATIs vs ngKATIs as well as in other compounds in their dataset? Is there a correlation between chemotypes and nuisance behavior?

Author response: We have provided additional detail on these interference chemotypes in our revised manuscript. The main text now states: “Other recently reported ngKATIs likely behave similarly. For example, the KAT7 inhibitor WM-3835 contains the same acylsulfonohydrazide scaffold as 470-472.³⁹ Neither WM-3835 or CPI-1612 (a KAT3 inhibitor) contain red-flag interference chemotypes found in many hKATIs (e.g., quinones, polyphenols).^{32,40}” We have also included a reference to the key paper describing the problematic chemotypes in the historical KAT inhibitors (ref 32).

Additional work is needed to more firmly correlate chemotypes with nuisance behavior in cellular assays, but in general we have observed highly electrophilic species like quinones leading to gross cytotoxicity. We have therefore added the following sentence in the discussion to speculate about this trend, and the need for additional studies to make more specific claims about chemotypes: “Future efforts could focus on the association between specific chemotypes and CP profiles, as well as phenotypic profiles in general. In this work, potent electrophiles (quinones, benzothiophene 1,1-dioxides, unstable succinimides, maleimides, etc.) produced strong CP profiles associated with cellular injury, whereas relatively weaker electrophiles (acrylamides) occasionally produced similar profiles at higher micromolar concentrations. Given our previous experiences determining structure-interference relationships with problematic chemotypes in biochemical assays², the generalization of chemotypes with specific CP profiles would benefit from testing a variety of analogs with and without the suspected problematic structural feature.”

Line 208: “The CP activities and relative cell numbers of 254 profiled compounds...”: Where does this number of compounds come from? Prior to this, the authors were describing a 218 compound dataset.

Author response: This 254 is composed of the 218 compound dataset, plus the addition of KAT inhibitors and targeted/nonspecific electrophiles. This also takes into account sample availability and assay throughput factors.

We have changed the text in our revised manuscript to indicate this: “The CP activities and relative cell numbers of 254 profiled compounds (218 cellular injury compounds plus KATIs and electrophiles, based on sample availability and assay throughput) were correlated with culture confluence (phase contrast), caspase-3/7 activation (GFP

channel, fluorogenic caspase 3/7 substrate), and cell viability (RFP channel, CytoTox dye which marks compromised membrane integrity) by live-cell imaging (Figure 4a)."

REVIEWER #4

In Dahlin et al., a generalizable cellular imaging approach and resource are provided for flagging nuisance compounds and prioritizing safer chemotypes from phenotypic discovery. The work is accompanied by publicly available cellular profiling images in U2OS. The authors studied 218 compounds in dose response by Cell Painting, a well-established technique from the same authors, and also compared the results to a related MLI dataset. Good correlation of the 2 datasets was found for many of the MLI compounds, but not all. I don't recall seeing any speculation around the compounds that were not correlated, this may be useful to add for some compounds even if anecdotally for one or two. The authors note strong connection between cell death/depletion and the CP bioactivity score, as well as other key markers of cellular imaging in live profiling.

Important observations were detailed around the improved next gen KATIs vs historical compounds. Similarly, non-specific electrophiles fared worse than targeted covalent drugs, including glutathione alterations, although notably in some cases the compound target wasn't present in U2OS (KRAS G12C, BTK). This is important because, for example, ibrutinib is noted to have drug-induced liver injury in some patients, and such a generalized approach would miss more tissue specific tox effects of drugs. It may be especially of interest for drug R&D to develop CP protocols in hepatocytes, cardiomyocytes, and other cell types more connected to safety profiling downstream of lead characterization. 24 hr and 48 hr timepoint comparisons showed 24 hr time points may often be used for efficiency's sake.

Author response: We thank the Reviewer for this overall feedback and believe this is an accurate summary of the manuscript.

Regarding the MLI compounds whose "historical" data correlated with our "newly generated" cell injury phenotype – but were not bioactive upon re-testing: we did not speculate on this because it is likely multifactorial. We were unable to discern any clear SAR or chemotype explanation in the compounds that were tested. Another explanation could be changes in the compound samples themselves, but again there was also no clear connection with chemical vendor or other sample information (sample age, QC, etc.). We have now included the following brief explanation of this in our revised manuscript: "There was no clear trend in terms of chemical structure (chemotypes) or sample information (vendor, QC, age) for those MLI-HC compounds that were not bioactive upon re-testing."

We also note that the chemical structure and QC information is provided in our supplemental dataset for all the MLI compounds tested in our study, and it should therefore be possible for readers to investigate in more detail if desired.

In Fig 1c, I can't read the cellular features text due to the colored columns and small font when reading the PDF version with figures positioned vertical. If the journal is willing to present Fig 1 horizontally on the page, this could be ameliorated.

Author response: We have increased the font size describing the columns in Figure 1C so that is more readable.

Figures 5 c/d/e/f- please check that the corresponding text in the manuscript corresponds with these panels appropriately and that all panels are referenced.

Author response: We thank the Reviewer for alerting us to these errors. We have corrected these figure references and added the appropriate corresponding text in our revised manuscript.

Finally- the key output of this resource paper for me is the list of suggested compounds covering various tox mechanisms. While Fig 6 gives an exemplar approach, I assume this is provided in Supplemental XLS File 1. However, it wasn't clear to me if the authors intended Column F "Informer Set" as their recommended set for anyone intending to follow this up. There is also Column D "Cellular injury" which is nice too. But I would really like it stated somewhere what the recommended set is, given that's the crux of the paper.

Author response: We have included a reference to the formulation of the proposed cellular injury informer set in our revised manuscript. The text now states: "Based on our data and cumulative experience with HTS, we propose an informer set of control compounds to model cell injury phenotypes in HCS and other phenotypic assays including mechanism-based and nonspecific modes of gross cellular injury (Figure 6; Supplementary Data 1, column "Proposed informer set")."

We have also modified the column header in the revised supporting file to better indicate the compounds we propose as part of a cellular injury informer set ("Proposed informer set").

Furthermore, we have modified our description of the Supplementary materials to now state: "Supplementary Data 1: Key compound descriptors (categories, SMILES, purity, annotations) for study compounds and proposed cellular injury informer set (XLSX)."

EDITORIAL FEEDBACK

Please complete or update the following checklists to verify compliance with our research ethics and data reporting standards. Address all points on the checklist, revising your manuscript in response to the points if needed. The forms must be downloaded and completed in Adobe Reader rather than opened in a web browser. Each form must be uploaded as a Related Manuscript file at the time of resubmission.

Editorial policy checklist:

<https://www.nature.com/documents/nr-editorial-policy-checklist.pdf>

Reporting summary:

Author response: We have completed the Editorial policy checklist and have included it in our revised submission. We have also included an updated Reporting summary in our revised submission.

Your paper uses custom code/software. Please complete the following code and software submission checklist and make your code available for reviewer assessment, if you have not already done so. The code/software can be provided in a zip file with a readme.txt file or other instructions for installing and running the software. If appropriate, also provide example data and expected output. If you have any issues with the file upload, please let me know.

<https://www.nature.com/documents/nr-software-policy.pdf>

Author response: Our manuscript relies on open-source (CellProfiler, R) and commercially available software (GraphPad Prism, Adobe Illustrator). It does not utilize custom code. Therefore, we believe this checklist is not applicable.

All Nature Communications manuscripts must include a “Data Availability” section after the Methods section but before the References. If any of the data can only be shared on request or are subject to restrictions, please specify the reasons and explain how, when, and by whom the data can be accessed.

Author response: Our original manuscript already contained a Data Availability section. In our revised manuscript, we have moved this section after the Methods section but before the References, as instructed.

Please also include a “Code Availability” section after the “Data Availability” section. If the code can only be shared on request, please specify the reasons.

Author response: Our manuscript does not utilize custom code. Therefore, we believe this section is not applicable.

All novel microarray, DNA sequencing, RNA-seq or proteomic datasets must be deposited in a publicly accessible database, and accession codes and associated hyperlinks must be provided in the “Data Availability” section.

Author response: Our manuscript does contain novel microarray, DNA sequencing, RNA-seq, or proteomic datasets. Therefore, we believe this section is not applicable.

We strongly encourage you to deposit all new data associated with the paper in a persistent repository where they can be freely and enduringly accessed. We recommend submitting the data to discipline-specific and community-recognized repositories.

Author response: We agree with the importance of persistent data repositories, and have already deposited the relevant data in such repositories (FigShare, Image Data Resource) where they are publicly available without restriction.

As noted in our Data Availability section: “The following data are deposited at Figshare ([10.6084/m9.figshare.20293992](https://doi.org/10.6084/m9.figshare.20293992)) and are available without restriction: (1) CP extracted features, (2) processed live-cell imaging data, (3) processed intracellular glutathione data, and (4) ALARM NMR spectra and UPLC-MS chromatograms for KAT inhibitors. The multi-terabyte collection of CP images, metadata, and associated CellProfiler object-level files are deposited at the Image Data Resource (idr.openmicroscopy.org, accession number idr0133).”

To maximise the reproducibility of research data, we strongly encourage you to provide a file containing the raw data underlying the following types of display items:

- Any reported means/averages in box plots, bar charts, and tables
- Dot plots/scatter plots, especially when there are overlapping points
- Line graphs

The data should be provided in a single Excel file with data for each figure/table in a separate sheet, or in multiple labelled files within a zipped folder. Name this file or folder ‘Source Data’, and include a brief description in your cover letter. The “Data Availability” section should also include

the statement “Source data are provided with this paper.”

Author response: We agree with the importance of reproducibility in research. That is why in our original and revised submission materials and associated links (Figshare, Image Data Resource), we have essentially included all of the relevant raw and processed data used for all the manuscript display items. In addition, as noted in our data availability statement, all data is available from the corresponding authors without restriction.

We also mandate the presentation of uncropped versions of any gels or blots, labelled with the relevant panel and identifying information such as the antibody used.

Author response: We have labelled the uncropped versions of our western blots in Supplementary Figure 6A to now specify the antibody used. We have also indicated the links between the cropped and uncropped blots by color coding the borders in our revised Supplementary Figure 6. Any additional uncropped gels are, of course, available upon request.

Please replace your bar graphs with plots that feature information about the distribution of the underlying data. All data points should be shown for plots with a sample size less than 10. For larger sample sizes, please consider box-and-whisker or violin plots as alternatives. Measures of centrality, dispersion and/or error bars should be plotted and described in the figure legend.

Author response: We have replaced the bar graphs in Supplementary Figure 5 with individual replicates as requested. We have noted measures of centrality and error bars in the corresponding figure legend.

Nature Communications is committed to improving transparency in authorship. As part of our efforts in this direction, we are now requesting that all authors identified as ‘corresponding author’ create and link their Open Researcher and Contributor Identifier (ORCID) with their account on the Manuscript Tracking System prior to acceptance. ORCID helps the scientific community achieve unambiguous attribution of all scholarly contributions. You can create and link your ORCID from the home page of the Manuscript Tracking System by clicking on ‘Modify my Springer Nature account’ and following these instructions. Please also inform all co-authors that they can add their ORCIDs to their accounts and that they must do so prior to acceptance.

Author response: Both corresponding authors (JLD, BKW) have our ORCID associated with our Springer Nature accounts. We have also informed all co-authors about the need to add their ORCIDs to their accounts prior to acceptance.

Thank you for your consideration of our revised manuscript.

Sincerely,

Bridget K. Wagner, Ph.D.

Jayme L. Dahlin, M.D., Ph.D.

Encl.

Reviewers' Comments:

Reviewer #1:

Remarks to the Author:

The authors have adequately addressed all topics previously noted by this reviewer. Revisions to the manuscript made in response to other reviewers have also improved the overall quality of the manuscript, most notably by including additional information in the figure captions and more carefully delineating how the results of the study support their conclusions.

Reviewer #2:

Remarks to the Author:

In the revised manuscript, the authors have addressed our primary comments. I have the following final comments:

The authors should write their reasoning for choosing the 9 clusters in the methods section of the paper. As described in the rebuttal the clustering, which forms the basis for the remaining of the paper, is done "manually", based on the researcher previous experiences. I don't think this is necessarily wrong, but it is important that this potential bias is made clear, at least in the methods section.

The correlation matrix in Fig S1 could be a useful visual to see the clusters, but without any annotations it is very hard to make use of. Even though it is aggregated profiles, could the relationship to the clustering used in Figure 1B perhaps be indicated, to show that this correlation matrix actually provides useful information? There are clearly clusters in the correlation matrix, but it is more or less useless without any annotations.

Reviewer #3:

Remarks to the Author:

The revised manuscript has addressed my critique adequately; thank you for careful revision.

Reviewer #4:

Remarks to the Author:

The authors have addressed all of my suggestions sufficiently.

The following are our responses to reviewer feedback:

REVIEWER #1

The authors have adequately addressed all topics previously noted by this reviewer. Revisions to the manuscript made in response to other reviewers have also improved the overall quality of the manuscript, most notably by including additional information in the figure captions and more carefully delineating how the results of the study support their conclusions.

Author response: We thank the Reviewer for their helpful comments which have greatly improved the quality of our revised manuscript.

REVIEWER #2

In the revised manuscript, the authors have addressed our primary comments. I have the following final comments:

The authors should write their reasoning for choosing the 9 clusters in the methods section of the paper. As described in the rebuttal the clustering, which forms the basis for the remaining of the paper, is done “manually”, based on the researcher previous experiences. I don’t think this is necessarily wrong, but it is important that this potential bias is made clear, at least in the methods section.

Author response: We thank the Reviewer for pointing out this omission on our part. The choice of nine clusters was done “manually” and based on our previous experiences. This includes providing a clear cluster for the microtubule compounds, which historically has been the most reproducible and strong phenotype in our hands. We also limited the cluster to nine, as more clusters tended to increasingly depend on apparent batch effects.

We have revised the text in our manuscript to now state: “The number of clusters was based manually on previous experiences to optimize the quality of the tubulin compound cluster, historically the most robust phenotype in previous experiments.”

The correlation matrix in Fig S1 could be a useful visual to see the clusters, but without any annotations it is very hard to make use of. Even though it is aggregated profiles, could the relationship to the clustering used in Figure 1B perhaps be indicated, to show that this correlation matrix actually provides useful information? There are clearly clusters in the correlation matrix, but it is more or less useless without any annotations.

Author response: We have updated Supplementary Figure 1 to now include the relationship with clusters 1-9. We calculated the correlation of each of the 216 (averaged) compound profiles against the (averaged) profiles for clusters 1-9, and aligned this to the right of the square heatmap. Therefore, it is now possible to visually inspect which of the nine clusters the compounds (and clusters) in the square heatmap most closely resemble.

Our revised legend now states: Supplementary Figure 1. Summary of cell painting compound clustering. Hierarchical clustering was performed on compounds (**1-171, 455-501**) using as their distance $1 - x$, where x is the pairwise Pearson correlation coefficient between compound treatments. The square heatmap depicts these pairwise correlation coefficients and is sorted according to the hierarchical clustering. The accompanying non-clustered heatmap to the right depicts the Pearson correlation coefficient of these same compounds against the consensus profiles for the 9 clusters. Note that the correlation coefficients were calculated using the average profile for each compound across all treatment concentrations. Data are from 24 h compound treatment times.

REVIEWER #3

The revised manuscript has addressed my critique adequately; thank you for cerful revision.

Author response: We thank the Reviewer for their helpful comments which have greatly improved the quality of our revised manuscript.

REVIEWER #4

The authors have addressed all of my suggestions sufficiently.

Author response: We thank the Reviewer for their helpful comments which have greatly improved the quality of our revised manuscript.

Thank you for your consideration of our revised manuscript.

Sincerely,

Bridget K. Wagner, Ph.D.

Jayme L. Dahlin, M.D., Ph.D.

Encl.